# Structure-Dependent Mechanism of Organophosphate Release from Albumin and Butyrylcholinesterase Adducts When Exposed to Fluoride Ion: A Comprehensive In Silico Study

**DOI:** 10.3390/ijms241914819

**Published:** 2023-10-01

**Authors:** Daria A. Belinskaia, Nadezhda L. Koryagina, Nikolay V. Goncharov, Elena I. Savelieva

**Affiliations:** 1Sechenov Institute of Evolutionary Physiology and Biochemistry, Russian Academy of Sciences, Thorez 44, 194223 St. Petersburg, Russia; 2Research Institute of Hygiene, Occupational Pathology and Human Ecology, Bld.93 p.o. Kuz’molovsky, 188663 St. Petersburg, Russia

**Keywords:** organophosphorus adducts, albumin, butyrylcholinesterase, biomarker, reactivation, fluoride ion, molecular docking, molecular dynamics, semiempirical methods of quantum chemistry

## Abstract

The most favorable targets for retrospectively determining human exposure to organophosphorus pesticides, insecticides, retardants, and other industrial organophosphates (OPs) are adducts of OPs with blood plasma butyrylcholinesterase (BChE) and human serum albumin (HSA). One of the methods for determining OP exposure is the reactivation of modified BChE using a concentrated solution of KF in an acidic medium. It is known that under the action of fluoride ion, OPs or their fluoroanhydrides can be released not only from BChE adducts but also from the adducts with albumin; however, the contribution of albumin to the total pool of released OPs after plasma treatment with KF has not yet been studied. The efficiency of OP release can be affected by many factors associated with the experimental technique, but first, the structure of the adduct must be taken into account. We report a comparative analysis of the structure and conformation of organophosphorus adducts on HSA and BChE using molecular modeling methods and the mechanism of OP release after fluoride ion exposure. The conformational analysis of the organophosphorus adducts on HSA and BChE was performed, and the interaction of fluoride ions with modified proteins was studied by molecular dynamics simulation. The geometric and energy characteristics of the studied adducts and their complexes with fluoride ion were calculated using molecular mechanics and semiempirical approaches. The structural features of modified HSA and BChE that can affect the efficiency of OP release after fluoride ion exposure were revealed. Using the proposed approach, the expediency of using KF for establishing exposure to different OPs, depending on their structure, can be assessed.

## 1. Introduction

Organophosphates (OPs) are widely used in agriculture and industry as pesticides, insecticides, plasticizers and flame retardants, polymeric materials, and components in the synthesis of drugs [1]. The toxic mechanism of OPs is mainly due to the irreversible inhibition of acetylcholinesterase (AChE, EC 3.1.1.7) of neuromuscular and neuronal synapses via covalent bond formation with catalytic Ser203 [2] (Figure 1).

However, AChE is not the only target of OPs in the human organism. Butyrylcholinesterase (BChE, EC 3.1.1.8) has an active site similar to AChE, viz the catalytic triad (Ser198, His438, and Glu325), the oxyanion center (Gly116, Gly117, and Ala199, which bind the carboxylic oxygen of substrates and the phosphoryl oxygen of OPs), and the anionic center (includes Trp82 and binds ammonium cationic groups of substrates and inhibitors) [3]. The same specific adducts are formed with BChE as with AChE. Moreover, some OPs form adducts with human serum albumin (HSA) [4,5,6], which is extremely important to consider in analytical studies. There are three main sites of interaction of HSA with low-molecular ligands (Sudlow I, Sudlow II, and site III), seven sites of interaction with fatty acids (FA1-7), and several secondary sites. Several sites of irreversible covalent binding of OPs to HSA have been described, but the hydroxyl group of Tyr411 in Sudlow site II is the most reactive [7,8,9,10].

Poisoning by OPs takes one of the leading places in the total number of exotoxicoses [11]. Thus, Perwitasari et al. [12] revealed that the majority of the surveyed farmers who used organophosphate pesticides showed symptoms of OP poisoning. In the study of Hardos et al. [13], the adducts of organophosphate ester flame retardants (OPE FR) with cholinesterases (ChE) were found in the blood of a quarter of the surveyed aircraft maintenance workers. Therefore, the retrospective establishment of OP poisoning based on the analysis of the formed adducts with HSA and ChE is an urgent task.

The limitations of modified AChE as a biomarker of OP poisoning are its relatively low concentration in plasma (0.5 μg/mL) and the difficulty of extracting it from erythrocyte membranes for analytical studies. BChE is freely available in serum/plasma at higher concentrations (5 µg/mL), making BChE a more suitable diagnostic biomarker. According to Degenhardt et al. [14], analysis is possible even if only 1% of BChE is inhibited. OPs react with albumin sites of modification at least 500 times slower than with ChE active sites [15,16]. However, the concentration of HSA in the blood is four orders of magnitude higher than that of BChE, so modified HSA is also used as a biomarker of OP exposure. Therefore, the main objects of the studies in the analytical toxicology of OPs are HSA and BChE. One of the methods for establishing OP poisoning is the reactivation of modified BChE using a concentrated solution of KF in an acidic medium (pH 3.5) [16,17,18,19,20]. As the result of the reaction of KF with the modified catalytic serine, OPs or their fluoroanhydrides are formed and are available for analysis. The scheme of BChE reactivation is shown in Figure 2. OPs or their fluoroanhydrides can be released not only from the adducts with BChE but also from the adducts with albumin, though some organophosphorus adducts on HSA could not be detected by fluoride exposure [21].

It is known that OP release from HSA and BChE adducts after fluoride ion exposure occurs with different efficiency, which is poorly reproduced in different experiments [16,17,18,20]. The efficiency of OP release can be affected by many factors associated with the experimental technique: losses during the extraction of analytes from plasma; the time from exposure of plasma to analysis (aging of adducts); the activity of blood enzymes due to both genetic and purely technical factors (for example, duration and conditions of plasma storage prior to use). The structure of the adduct is one of the main factors determining the efficiency of reactivation. However, none of the available literature sources have performed the comparative structural and functional analysis, which could reveal how the chemical structure of the adduct can affect the efficiency of regeneration of various plasma proteins with fluoride ion. Such information could expand our knowledge about the plasma proteins’ reactivation mechanism, improve the methodology for detecting OP poisoning, and provide the possibility of quantitative analysis and the appointment of adequate therapy for patients who survived acute, subacute, or chronic poisoning.

The purpose of the work reported on here was to perform a comparative analysis of the structure and conformation of organophosphorus adducts on HSA and BChE using molecular modeling methods, study the mechanism of adduct cleavage by fluoride ion, identify structural features of adducts that can affect the efficiency of this process, and evaluate the prospects for creating a technology for a preliminary computer evaluation of the efficiency of OP release from HSA and BChE adducts as one of the procedures for establishing the exposure to OPs.

## 2. Results

### 2.1. Selection of Adducts for In Silico Investigation Based on the Available Experimental Data on Their Structure and Features

The adducts for investigation were selected primarily on the basis of the availability of known experimental data on their structure and properties, which makes it possible to compare the results of molecular modeling with the results of in vitro experiments. Therefore, different sets of adducts were chosen for BChE and HSA.

The structures of the studied BChE adducts (Ad1–Ad11) are shown in Table 1. Ad1–Ad10 are described by the general formula given in the table. These adducts were selected because they are the products of the interaction of known OPs with BChE [22]. For example, adducts Ad2 and Ad3 are formed as the result of exposure to paraoxon and armine, respectively. However, adduct Ad2 can also be formed by the reaction of BChE with insecticide diethylfluorophosphate or with flame retardant triethylphosphate since paraoxon, diethylfluorophosphate, and triethylphosphate have a different leaving group X (Figure 1), but the same substituents R1 and R2. With the exception of Ad2, for all studied adducts, the phosphorus atom is a chiral center. We studied only those stereoisomers of the phosphorus group described by the general formula in Table 1 because these stereoisomers of OPs are more active against human cholinesterases [23,24]. Adducts Ad9 and Ad10 are stereoisomers at the chiral carbon atom (Table 1). It is known that organophosphorus adducts of BChE undergo dealkylation (aging) over time, especially Ad9 and Ad10. However, in our work, we studied only nonaged adducts of BChE.

Adduct Ad11 deserves special mention. It is the product of the interaction of BChE with 4H-1,3,2-benzodioxaphosphorin, 2-(2-methylphenoxy)-, 2-oxide (CBDP), which in turn is the product of in vivo metabolism of well-known flame-retardant tri-ortho-cresyl phosphate. For adduct Ad11, we also studied only the nonaged form. There is the experimentally obtained three-dimensional structure of the nonaged adduct of CBDP with AChE in the protein databank (PDB) (entry 4bc1 [25], accessed on 1 September 2023), which we used as a comparison structure in conformational analysis.

The structures of the studied HSA adducts are shown in Table 2.

Adduct Ad12 is an albumin analogue of adduct Ad2 (Table 1). We chose this adduct for two reasons. First, Ad12 is a product of the interaction of HSA with paraoxon, which has been widely studied [8,26,27,28]. Like Ad2, Ad12 can be the product of the interaction of HSA with diethyl fluorophosphate, but not with the fire-retardant triethyl phosphate since, according to Chu et al. [29], triethyl phosphate (like other OPE FR) does not form adducts with albumin. The second reason is the simplicity of the chemical structure of Ad12. Ad12 has no phosphorus stereoisomerism. While the stereoselectivity of the ChE active site is well studied, and we can afford to study adducts of only active stereoisomers of OPs, there is no such experimental background in the case of HSA. Therefore, it is convenient to use Ad12 at the initial stage of studies devoted to the conformational analysis of albumin adducts and the possibility of their cleavage by fluoride ion.

Adducts Ad13 and Ad14 are albumin analogues of adduct Ad1 (Table 1). They are stereoisomers at the phosphorus atom relative to each other. As noted above, the stereoselectivity of albumin is poorly studied, so, unlike BChE, we investigated both enantiomers. Adduct Ad13 and its stereoisomer Ad14 were chosen because this pair of adducts was experimentally shown to be capable of cleavage by fluoride ion [21].

Theoretically, Ad12–Ad14 adducts can be formed not only with tyrosines (the most reactive of which is Tyr411) but also with serines, threonines, and lysines. However, John et al. [8] studied HSA adducts with organophosphates (including paraoxon) by MALDI-TOF MS and showed that the main adducts are formed on Tyr411, Tyr150, and Tyr161. Therefore, we included only tyrosine in the general scheme describing the structure of Ad11–Ad13 albumin adducts. Moreover, it is interesting to note that John et al. [8] did not reveal aged (dealkylated) HSA adducts for the studied set of OPs; therefore, for adducts Ad12–Ad14, we considered only nonaged forms.

In the final stage, we studied HSA adducts with CBDP. As noted, OPE FR cannot directly interact with albumin and form adducts with this protein [29]. However, it is known that CBDP, which is a metabolite of tri-ortho-cresyl phosphate, can interact with HSA [30,31]. According to [30,31], CBDP can form adducts with several tyrosine, lysine, and histidine residues of HSA. Their structural formulas are presented in Table 2 (adducts Ad15–Ad17). According to the interaction scheme between CBDP and HSA proposed in [30,31], the ring-open adduct is formed at the first stage of the CBDP interaction with tyrosines, which then loses the saligenin group and turns into *o*-cresyl phosphate. When CPDP interacts with tyrosines of HSA, only *o*-cresyl phosphate adducts were detected [30,31]. Therefore, unlike Ad11 (which is the nonaged adduct of CBDP with BChE), Ad15 (tyrosine adduct of CBDP with HSA) is dealkylated (Table 2). When CBDP interacts with lysines and histidines, due to the specifics of the nucleophilic attack (on carbon but not phosphorus atom), a dealkylated ring-opened adduct is formed at the first step of the reaction. Moreover, the histidine adduct then loses the phosphate moiety [31]. For this reason, Ad17 cannot be called an organophosphorus adduct, and formally, it falls outside the scope of this article. However, we included the modeling of Ad17 in our work to explore the conformational features of the full range of CBDP-HSA adducts described in [31].

### 2.2. Conformational Analysis of Three-Dimensional Models of Organophosphorus Adducts on BChE

The crystal structures of Ad1 and Ad4 adducts with human BChE can be found in the PDB database [32] (entries 3djy [33] and 2xqk [34], respectively, accessed on 1 September 2023). These structures (with hydrogen atoms added by us) are shown in Figure 3A and Figure 3B, respectively. These structures were used to build 3D models of other BChE adducts. The procedure is described in more detail in Section 5.1.

Three-dimensional models of BChE adducts (obtained by molecular docking with further optimization by the energy minimization method) correspond to known experimental data on the conformation of OP adducts (phosphoryl oxygen is bound in the oxyanion center, and His438 hydrogen interacts with the oxygen of the *O*-alkyl substituent). Figure 4 shows the obtained three-dimensional structures of Ad2 (Figure 4A) and Ad11 (Figure 4B) adducts as examples. It can also be noted that one of the aromatic rings of adduct Ad11 forms a Y-shaped pi–pi interaction with the side chain of Trp82. The topology of the modified BChE, which forms the entrance to the active site gorge and includes the location of the active site relative to the peripheral anionic site, is presented in Appendix A.

### 2.3. Conformational Analysis of Three-Dimensional Models of Organophosphorus Adducts on HSA

As noted, John et al. [8] identified three main interaction sites of paraoxon and other OPs with human albumin: Tyr150, Tyr161, and Tyr411. Using the same approach that we used to build adducts of OPs with BChE, we prepared the three-dimensional models of adduct Ad12 with Tyr150, Tyr161, and Tyr411 (Ad12-Tyr150, Ad12-Tyr161, and Ad12-Tyr411, respectively). The procedure is described in more detail in Section 5.1. Since there are no experimental data on the conformation of adducts Ad12-Tyr150, Ad12-Tyr161 and Ad12-Tyr411, the models built by molecular docking method were optimized not only by the energy minimization (as in the case of BChE) but additionally by 50 ns molecular dynamics (MD) simulation. The stable conformations of adducts Ad12-Tyr150, Ad12-Tyr161, and Ad12-Tyr411 are shown in Figure 5.

According to the obtained data, in adduct Ad12-Tyr150 (Figure 5A), the imidazole group of His242 interacts with the phosphoryl oxygen atom of the adduct (His242 plays the role of the oxyanion center), and the guanidine group of Arg257 interacts with one of the etheric oxygens of the OP moiety (similar to His438 of BChE). All three amino acids (Tyr150, His242, and Arg257) belong to site Sudlow I [36] (Appendix A).

In adduct Ad12-Tyr161, the aromatic ring of Tyr161 forms parallel pi–pi interaction with the aromatic ring of Tyr138 and T-shaped pi–pi interaction with Phe165 (Figure 5B). The phosphoryl oxygen atom of the adduct does not form any close interactions with the surrounding amino acids. In adduct Ad12-Tyr411, the NH_3_-group of Lys414 interacts with atom Oη of the modified Tyr411 (Figure 5C) and the phosphoryl oxygen atom of the adduct. Most amino acids interacting with adduct Ad12-Tyr161 belong to site III (the same binding site that binds bilirubin and its derivatives [37], Appendix A).

Tyr411 of HSA and neighboring Lys414 and Arg410 belong to site Sudlow II [36] (Appendix A). As in the case of Ad12-Tyr161, adduct Ad12-Tyr411 does not form any close interactions with the surrounding amino acids. Thus, according to molecular modeling data, modified Tyr150 and auxiliary amino acids His242 and Arg257 most closely resemble the structure of the active site of modified BChE.

### 2.4. Formation of CBDP Adducts with HSA According to Molecular Modeling Data

According to Schopfer et al. and Liyasova et al. [30,31], CBDP can form adducts with the following amino acid residues of HSA: His67, His146, His242, His247, His338, Tyr138, Tyr140, Lys199, Lys351, Tyr411, Lys414, Lys432, and Lys525. The schemes of the formation of CBDP adducts with tyrosines and histidines are shown in [31]. In the case of tyrosines, when the nucleophilic attack of the hydroxyl oxygen atom on the phosphorus atom of CBDP occurs, it results in the formation of the ring-opened adduct, which is then hydrolyzed to form *o*-cresyl phosphotyrosine. In the case of histidine, the nucleophilic attack of atom Nε on the carbon atom of the methylene group of CBDP results in the formation of the ring-opened adduct, which is further hydrolyzed to *o*-hydroxybenzyl adduct. During this reaction, atom Nδ of the imidazole ring of histidine is protonated, while atom Nε is deprotonated. Adducts with lysine are formed by a mechanism similar to histidine’s.

To find CBDP adducts with HSA, the authors of [30,31] used a saturating concentration of CBDP several times higher than the concentration of albumin. At lower concentrations, CBDP interacts with the most preferred sites in the HSA molecule. To determine these sites, we studied the interaction of CBDP with HSA using molecular modeling methods.

#### 2.4.1. Molecular Docking of CBDP to HSA Sites of Adducts Formation

We performed molecular docking of CBDP into the sites of its interaction with albumin. Among the conformations found, we selected only the productive conformations, namely those in which the distance between the functionally significant atoms (oxygen/nitrogen atom of catalytically active amino acids and phosphorus/carbon atom of CBDP) does not exceed 4 Å so that the nucleophilic attack described above can occur. The geometric and energy characteristics of the found productive conformations are presented in Table 3.

According to the obtained data, the most preferred (primary) interaction sites between CBDP and albumin are amino acids His242, Lys351, Tyr411, and Lys414. His242 is located in Sudlow site I, Tyr411 and Lys414 are located in Sudlow site II, and Lys351 is located in fatty acid binding site FA6. Productive conformations of CBDP near Tyr138 and Lys432 were not found. It appears that the hydrolysis of CBDP at these sites is controlled by allosteric modulation; the interaction of CBDP with the main sites of albumin leads to conformational changes in the protein globule, and only after these changes, the formation of adducts with Tyr138 and Lys432 is possible.

As mentioned above, the authors of [31] used a saturating concentration of CBDP several times higher than the concentration of albumin to find CBDP adducts with HSA. At lower concentrations, CBDP will interact mostly with the primary sites His242, Lys351, Tyr411, and Lys414. Therefore, we selected only CBDP complexes with sites His242, Lys351, Tyr411, and Lys414 for further study and did not consider other sites.

It is interesting to compare the obtained ΔG values with known experimental data. For the highest affinity binding sites (His242, Lys351, Tyr411, and Lys414), the estimated ΔG values are close to −7 kcal/mol, which at room temperature corresponds to the value of dissociation constant (K_d_) of about 7 μM. We have not found experimentally obtained K_d_ or ΔG values for the interaction of CBDP with HSA in the literature, so we can only rely on other OPs. For the in vitro interaction of paraoxon with site Sudlow II of HSA, we obtained the Kd value of 4 μM [27]. Dahiya et al. [38] studied the interactions of chlorpyrifos, diazinon, and parathion with bovine serum albumin (BSA). The obtained K_d_ values for these OPs were 1.81 × 10^−4^ M, 1.30 × 10^−3^ M, and 1.11 × 10^−3^ M, respectively. Čadež et al. [39] studied the interactions of several OPs with AChE and BChE; the experimentally obtained K_d_ values varied in the range of about 4 to 100 μM. It is logical to assume that the efficiency of interaction of these compounds with albumin is weaker since ChEs are the main targets of the OPs’ action. Thus, the values of ΔG obtained in silico tend to overestimate interaction efficiency because of the limitations of molecular modeling methods. Therefore, when analyzing in silico data, it is relevant not to analyze absolute ΔG values but only make comparisons in effectiveness between substances and binding sites.

#### 2.4.2. Molecular Dynamics of the CBDP Complexes with the Main Sites of Its Interaction with HSA

The conformational changes in CBDP complexes with sites His242, Lys351, Tyr411, and Lys414 of HSA were modeled by 100 ns MD simulation. Using the obtained trajectories, we calculated the time dependence of the distance between the functionally significant atoms (oxygen/nitrogen of the catalytically active amino acids and phosphorus/carbon of CBDP). The plots of these dependencies are shown in Figure 6. Figure 7 shows the stable conformations of CBDP within the studied sites.

The position of CBDP near His242 is stable. Throughout the simulation, the distance between the nitrogen atom Nε of His242 and the carbon atom of the methylene group of CBDP did not exceed 4 Å (Figure 6A), which makes possible the nucleophilic attack of the nitrogen atom on the carbon atom of CBDP (Figure 7A). The stability of the productive conformation indicates a high reactivity of His242 towards CBDP and a high rate of adduct formation.

The position of CBDP near Lys351 is unstable. After energy minimization and equilibration, the molecule of CBDP moves away from Lys351 by more than 5 Å and does not approach closer than 5 Å during the entire simulation (Figure 6B). However, the molecule of CBDP remains in site FA6, forming a stable hydrogen bond with the side chains of Arg209 and Lys323 (Figure 7B).

The position of CBDP near Tyr411 is stable for a part of the simulation. During the first 4 ns of the simulation, the distance between the phosphorus atom of CBDP and the hydroxyl oxygen atom of Tyr411 does not exceed 4 Å, but then the ligand molecule moves away from the tyrosine (Figure 6C). In the productive conformation, the molecule of CBDP is fixed by the hydrogen bond between the phosphoryl oxygen atom of the ligand and the hydrogen atom of the hydroxyl group of Tyr411, as well as by pi–cation interaction between the guanidine group of Arg410 and the aromatic ring of CBDP. The proximity of CBDP and Tyr411 allows the nucleophilic attack of the tyrosine hydroxyl on the phosphorus atom of CBDP (Figure 7C).

The position of CBDP near Lys414 is unstable. After energy minimization and equilibration, the molecule of CBDP moves away from Lys414 by more than 5 Å and does not approach closer than 6 Å during the entire simulation (Figure 6D). However, the molecule of CBDP remains in Sudlow site II, forming a stable hydrogen bond with the side chain of Arg410 (Figure 7D).

Thus, according to molecular modeling data, CBDP primarily forms adducts with His242 and Tyr411 in the sites Sudlow I and Sudlow II, respectively.

#### 2.4.3. Conformational Analysis of CBDP Adducts with HSA

According to the calculations described in Section 2.4.2, CBDP preferentially interacts with Tyr411 and His242 sites. At the next stage, we built models of CBDP adducts with Tyr411 and His242 of HSA (Ad15 and Ad17) and applied 50 ns MD simulation to model the conformational changes in HSA molecule after the formation of these adducts. Additionally, we performed a similar study for CBDP adduct with Lys414 (Ad16). According to the data described in Section 2.4.2, CBDP does not form a stable, productive conformation in the Lys414 site, but according to molecular docking, Lys414 is most potent for adduct formation among all HSA lysine found in [31] (Table 3).

The conformations of adducts Ad15–Ad17 and their nearest amino acid environment after 50 ns of simulation are shown in Figure 8. In adduct Ad15 with Tyr411, one of the phosphoryl oxygens of *o*-cresylphosphotyrosine forms a hydrogen bond with the hydroxyl group of Ser489, the second phosphoryl oxygen forms the salt bridge with the hydrogen atom of the sidechain of Arg410 (Figure 8A). Another hydrogen of the guanidine group of Arg410 forms the hydrogen bond with the etheric oxygen of the adduct. The aromatic ring of the adduct is localized around residues Leu387, Asn391, and Leu453. In adduct Ad16 with Lys414, one of the phosphoryl oxygens forms the hydrogen bond with the hydroxyl group of Ser489, and the second one forms the salt bridge with the hydrogen atom of the sidechain of Arg410 (Figure 8B). The etheric oxygens of the adduct interact with the hydroxyl group of Tyr411 and the sidechain of Arg410. In adduct Ad17 with His242, an *o*-hydroxybenzyl adduct is localized near residues Tyr150, Arg257, and Gln196 (Figure 8C), forming pi–cation interaction with the guanidine group of Arg257.

Interestingly, the conformation of adduct Ad12-Tyr411 (nonaged paraoxon adduct with Tyr411, Figure 5C) is not similar to that of dealkylated adduct Ad15. In the case of Ad15, Arg410 and Ser489 serve as the oxyanion center, binding the phosphoryl oxygens of the adduct (Figure 8A). In the case of Ad12-Tyr411, however, the phosphoryl oxygen of the adduct does not form any close interactions with the surrounding amino acids. Thus, the conformations of HSA adducts can differ significantly depending on the structure of the substituents and the state of alkylation. This feature of HSA differs from BChE, where the conformations of the adducts are generally similar to each other (the phosphoryl atom is bound in the oxyanion center formed by Gly116, Gly117, and Ala199, while His438 interacts with the oxygen of the *O*-alkyl substituent).

### 2.5. Docking of Fluoride Ion into the Active Site of Modified BChE

At the next stage, we evaluated how the structural features of BChE adducts can affect the efficiency of reactivation of modified BChE with fluoride ion. The process of reactivation of BChE with fluoride ion consists of the formation of the new covalent bond between the fluorine atom and the phosphorus atom of the adduct, with simultaneous destruction of the covalent bond between the phosphorus atom and the hydroxyl oxygen of the catalytic serine of inhibited BChE. Geometrically, the nucleophilic attack of the fluoride ion is possible from four directions according to the number of trihedral angles in the tetrahedral structure formed by the phosphorus and neighboring atoms [40] (Figure 3 and Figure 4). It is logical to assume that the nucleophilic attack will occur more likely from the direction where the position of the fluoride ion could be stabilized, for example, by a hydrogen bond. The analysis of the amino acid environment of the catalytic triad revealed only one possible candidate for the role of a “stabilizer”, namely His438, which can hold the fluoride ion in a productive conformation (a conformation in which the nucleophilic attack is possible). In this case, the nucleophilic attack occurs in the space of the trihedral angle formed by the phosphorus atoms, substituent O-R1, substituent R2, and the hydroxyl oxygen of the catalytic serine (Figure 3 and Figure 4). Therefore, in subsequent computational experiments, we proceeded with the assumption that the fluoride ion attacks the phosphorus atom from the direction of His438. In confirmation of the geometry we have chosen, we note the work of Lushchekina et al. [40], in which BChE reactivation by a water molecule was studied using molecular modeling methods. It was shown that the replacement of some amino acids from the environment of the catalytic triad with histidine residues lowered the energy barrier of the hydrolysis of diethylphosphorylated catalytic serine as a result of the stabilization of the water molecule by these “new” histidines.

The procedure for fluoride ion docking to the active site of inhibited BChE is described in detail in Section 5.2. An example of a three-dimensional model of the complex of inhibited BChE with fluoride ion is shown in Figure 9.

Further, the stability of these complexes was checked by MD simulation.

### 2.6. Molecular Dynamics Simulation of the Fluoride Ion and Inhibited BChE Complexes

Using MD simulation, we studied the stability of the position of the fluoride ion in the active site of inhibited BChE. Using the obtained trajectories, the time dependence of the distance between the fluoride ion and the phosphorus atom of the adducts (distP–F) was plotted (Figure 10).

Adducts Ad1 and Ad2 (Ad11 too, but we consider it separately) differ from other adducts by the presence of an electronegative atom in substituent R2 (nitrogen and oxygen, respectively). According to the obtained data, in adduct Ad1 (Figure 10A), the fluoride ion remains in the productive position for 1.3 ns, then moves away from the catalytic Ser198, but at the same time continues to interact with protonated His438. The fluorine ion behaves similarly in the case of adduct Ad2 (Figure 10B) and remains in the productive position even longer, for around 4 ns.

Adduct Ad3 differs from Ad2 by the absence of an oxygen atom in substituent R2, which affects the behavior of the fluoride ion in the BChE active site. The fluoride ion remains at a close distance from the phosphorus atom of adduct Ad3 for about 3 ns (Figure 10C). However, even visually, in Figure 10B,C, it can be seen that the fluctuation of the fluoride ion relative to its average position in the case of Ad3 is wider compared with Ad2. In adduct Ad4, the fluoride ion remains in the productive position for 2.2 ns, then His438 and the ion move away from Ser198, maintaining interaction with each other (Figure 10D). Interestingly, the shortening of substituent R2 by one carbon atom (relative to Ad3) slightly stabilizes the position of the fluoride ion; its fluctuation relative to the average position in the case of Ad4 is more limited compared with Ad3.

In the case of Ad5, the fluoride ion remains in the productive position for 3 ns (Figure 10E), which is almost 1 ns longer compared with Ad4. Then, the fluoride ion moves away from the catalytic serine and finally leaves the active site of BChE; the distance between the fluorine and phosphorus atoms is ~70 Å at the final step of the trajectory. The analysis of the movement of the fluoride ion showed that it leaves the enzyme along the gorge of the active site, passing by Asp70 and Tyr332 of the peripheral anionic site. Adduct Ad6 differs from Ad5 by one more carbon atom in substituent R1. In the case of Ad6, the fluoride ion remains in the productive conformation for 7 ns (Figure 10F), i.e., the extension of substituent R1 stabilizes the system in this case.

In the case of Ad7, the fluoride ion stays at a close distance (~4 Å) from the catalytic Ser198 for 2 ns, then it moves away but returns to the productive position again (at a distance of ~3.0–3.5 Å from the phosphorus atom), and then finally moves away from the catalytic triad (Figure 10G). In other words, the linearization of substituent R1 (relative to Ad6) reduces the stability of the system. In the case of the cyclohexane substituent R1 in adduct Ad8, the fluoride ion remains in the productive position for 1.4 ns (Figure 10H). Then, the fluorine–His438 pair moves away from the modified Ser198.

Adducts Ad9 and Ad10 have the most branched substituent R1 of all the studied compounds. According to the obtained data, in adduct Ad9, the fluoride ion remains in the productive conformation for 1.1 ns (Figure 10I). Then, the active site changes its conformation and remains thus until the end of the simulation. In this conformation, the fluoride ion still interacts with the imidazole ring of His438, but this pair moves away from the modified Ser198. In Ad10, the fluoride ion remains in the productive position for 8.5 ns (Figure 10J). Then, similarly to adduct Ad9, the fluorine–His438 pair moves away from the modified Ser198. In the active center of C*_R_*-stereoisomer (Ad10), in contrast to C*_S_*-stereoisomer (Ad9), the tert-butyl group is directed towards the fluorine–His438 pair, and thus the steric interactions keep the fluoride ion in the productive position longer.

The conformational behavior of adduct Ad11 (Figure 10K) is similar to Ad2 (Figure 10B) despite the significant difference in steric size between their substituents. In the case of Ad11, the fluoride ion also remains in the productive conformation for 4 ns, then moves away from the phosphorus atom.

Thus, for all adducts, except for Ad7, the complexes of inhibited BChE with the fluoride ions remain in the stable conformation during the first ns of simulation. In the case of Ad7, this complex is maximally stable in the interval 6.5–8.5 ns. To analyze the stable conformations, we chose the trajectory intervals 6.5–7.5 ns for adduct Ad7 and 0–1 ns for other adducts. The average distance between the phosphorus and fluorine atoms in each interval was calculated. Then, we extracted the coordinates of the systems from the molecular dynamics trajectories after 1 ns of simulation (after 7.5 ns for Ad7) and performed the conformational analysis of the extracted conformations. The analysis of the obtained data is presented in the following sections.

### 2.7. Conformational Analysis of the Fluoride Ion and Inhibited BChE Complexes

The stable, productive positions of the fluoride ion in the active site of modified BChE are shown in Figure 11. Of particular interest is the location of the fluoride ion relative to Ser198, His438, and nearby water molecules.

In adducts Ad1 and Ad2 with quite massive substituents R2, the fluoride ion is surrounded by two water molecules (Figure 11A,B). In the series of Ad3–Ad7, the fluoride ion is hydrated by three water molecules (Figure 11C–G), and in adduct Ad8 with cyclohexane substituent R1, by four H_2_O molecules (Figure 11H). It is interesting to note the stereoselectivity of fluoride ion hydration; adducts Ad9 and Ad10 are stereoisomers, and in the case of adduct Ad9, the fluoride ion is hydrated by four water molecules (Figure 11I), and in Ad10 by three (Figure 11J). In adduct Ad11 with the heaviest R2 substituent, the fluoride ion is hydrated with three water molecules (Figure 11K).

Thus, the trend can be noticed where, in the case of massive R2 substituents, there is space for only two water molecules in the active site of inhibited BChE. With the shortening of substituent R2 to an ethyl and methyl group, a place for the third water molecule appears in the active center. Adduct Ad11 is an exception due to its substituent R2 containing a polar hydroxyl group that forms an intramolecular hydrogen bond with oxygen atom Oγ of the modified Ser198 (Figure 11K). This hydrogen bond pulls the entire substituent R2 away from the fluoride ion and clears a place for the third water molecule.

In adduct Ad9 (Figure 11I), in contrast to Ad10 (Figure 11J), the massive tert-butyl group is directed away from His438, giving some space in the active site for an additional water molecule. The case of Ad7 is similar; although the adduct contains the massive cyclohexane group, this substituent is elongated in the “chair” conformation along the axis of the active site, occupying less space in the active site than we would expect if taking into account only the number of atoms in substituent R1.

As the steric size of substituent R1 gradually increases, the attraction between the adduct and the amino acids of the active center gradually increases (according to the formula describing van der Waals energy). However, when the steric size of the substituent reaches the critical value, repulsion begins, and the adduct tends to go into a conformation in which the massive substituent is far from the active site. The space inside is thus freed for the fourth water molecule.

The possibility of reactivation of inhibited Ser198 depends on the strength of bond P–Oγ and the ease of formation of new bond P–F. It is logical to assume that the stronger the fluorine ion is hydrated, the harder it forms a new covalent bond with the phosphorus atom of the adduct. Thus, in the very first approximation, based only on the degree of hydration of the fluoride ion, it is possible to divide adducts into three groups of relatively easy (Ad1 and Ad2), moderate (Ad3–Ad7 and Ad10–Ad11), and weak (Ad8 and Ad9) ability to be reactivated with fluoride ion.

### 2.8. Estimation of the Efficacy of BChE Reactivation with Fluoride Ion Using Molecular Mechanics Approach

To break bond P–Oγ and form a new bond P–F, the system must overcome the energy barrier, which is determined by the difference in the energies of bonds P–Oγ and P–F. In the molecular mechanics (MM) approach, in contrast to quantum mechanics, the molecule is considered not as a set of nuclei and electrons but as a system of indivisible atoms. In the MM, the energy of a covalent bond is determined only by the types of atoms that form the bond; therefore, the energy of bond P–Oγ is the same for all adducts. The probability of reactivation, in this case, is determined only by the energy of attraction between the phosphorus atom and the fluoride ion—the energies of electrostatic interaction and van der Waals forces. Since the charges in atoms do not depend on the conformation of the system in the classical MD method, the difference in the values of these energies for different adducts is determined only by the distance between the fluorine and phosphorus atoms in the productive conformation; the shorter this distance is, the easier the reactivation occurs. Table 4 presents the average distances between fluorine and phosphorus atoms during 1 ns of MD simulation (intervals 7.5–8.5 ns for Ad7 and 0–1 ns for the other adducts).

According to the given values, the studied adducts are ranked Ad1, Ad11, Ad2, Ad7, Ad4, Ad5, Ad6, Ad10, Ad3, Ad8, and Ad9, according to the degree of increasing complexity of reactivation. Expectedly, the greater the distance between the fluoride ion and the phosphorus atom of the adduct, the more strongly the fluorine ion is hydrated. As we assumed in the previous section, the efficiency of reactivation decreases with a decrease in the size of substituent R2 and with an increase in the size of substituent R1. Notable exceptions are the adducts Ad11, Ad7, and Ad3. We have already explained in the previous section how, because of the intramolecular hydrogen bond, substituent R2 of Ad11 occupies less space between Ser198 and His438 than substituent R2 of adduct Ad1. The exclusion of Ad7 from the general trend indicates that not only the number of carbon atoms in substituent R1 is important but also the branching of this fragment: the linear *n*-butyl group of Ad7 occupies as much space in the region of the catalytic triad as the ethyl group of adduct Ad4. An interesting exception is the adduct Ad3. Its substituent R2 is more massive than the one of Ad4 but less massive than the R2 of Ad1 and Ad2 (Table 4). Therefore, in the case of Ad3, the third water molecule can fit into the active site of BChE, but because of the greater hydrophobicity of substituent R2 compared with Ad1 and Ad2, the hydrated fluoride ion moves away from the phosphorus atom (Table 4).

### 2.9. Estimation of the Efficacy of BChE Reactivation with Fluoride Ion Using Semiempirical Approach

At the next stage, we recalculated the charges on atoms, the length of P–Oγ covalent bonds, and the distance between fluorine and phosphorus atoms using the Austin Model 1 (AM1) semiempirical method [41]. Reactivation of ChE catalytic serine modified by OPs (whether through spontaneous reactivation with a water molecule, reactivation with fluoride ion, or oximes) proceeds by forming an intermediate tetrahedral complex. A schematic representation of such a complex in the example of oximes is shown in Franjesevic et al. [42]. Analogously, we have schematically depicted a similar intermediate complex for the case of BChE reactivation with fluoride ion (Figure 12A). Then, with the help of the semiempirical method Austin Model 1 (AM1), we calculated the value of ΔE_el_, which is the difference between the values of E_el_P–Oγ and E_el_P–F, where E_el_P–Oγ and E_el_P–F are the energies of electrostatic interactions between the phosphorus atom of the adducts (P), the hydroxyl oxygen atom of Ser198 (Oγ), and the fluoride ion (F). The calculation technique is described in detail in Section 5.4. Figure 12B shows all of the described characteristics in the example of the complex of the fluoride ion with adduct Ad4.

The types of atoms involved in the reactivation are the same for different adducts. Moreover, the steric size of the fluoride ion is smaller compared with the organophosphorus moieties of the adducts and the residues of the active site. Therefore, the main energy contribution to the interaction between the fluoride ion and the surrounding amino acids is made by electrostatic interactions. For example, in the case of the productive complex of the fluoride ion with adduct Ad1, the energy of electrostatic interactions between the fluoride ion and modified BChE is −53.9 kcal/mol, and the energy of van der Waals forces is +5.9 kcal/mol in the MM approach. If we accept that the energies of covalent bonds P–O and P–F are the same for all adducts and products of their reactivation and neglect the difference in the values of the energy of van der Waals forces between the fluoride ion and the studied adducts, then the value of ΔE_el_ represents a simplified (requiring minimal computational time) model of the energy barrier that the system needs to overcome in order to break covalent bond P–Oγ and form new covalent bond P–F. The lower the value of ΔE_el_, the easier the adduct is reactivated. The electrostatic and geometric characteristics of the complexes of the fluoride ion with BChE adducts according to AM1 are presented in Table 5.

According to the calculated values of ΔE_el_, the studied adducts are ranked Ad7, Ad11, Ad1, Ad2, Ad6, Ad4, Ad3, Ad5, Ad8, Ad10, and Ad9 (Table 6), according to the degree of increasing complexity of reactivation.

Here, we examine all the adducts in detail. Adducts Ad1 and Ad2 can be placed into a separate group because of the structure of their substituent R2; in Ad1, substituent R2 is N-(CH_3_)_2_ group (the most branched R2 of all studied adducts), while R2 of adduct Ad2 is a massive O-ethyl moiety. Because of the large steric size of these substituents, the active site of BChE is smaller and less accessible to water molecules. Consequently, the fluoride ion is less hydrated in the active site compared with other adducts (Figure 11A), which allows it to come closer to the phosphorus atom (Table 4 and Table 5). In turn, it reduces the value of E_el_P– F and facilitates the formation of a new bond, P–F. On the other hand, in the case of Ad1 and Ad2, the fluoride ion in the BChE active site has one of the most negative charges and consequently has a negative E_el_P–F value (Table 5), which also contributes to a decrease in the energy barrier.

Adduct Ad11 stands apart from other adducts because of the intramolecular hydrogen bond between substituent R2 and atom Oγ of the modified Ser198 (Figure 11K). Because of this bond, first, the conformation of substituent R2 allows the fluoride ion to come closer to the phosphorus atom, and second, it increases the charge on the phosphorus atom. The value E_el_P–F in Ad11 is the most negative of all adducts; therefore, the final value of ΔE_el_ is rather low, and according to our calculations, the adduct is well reactivated by fluoride ion.

Adducts Ad5, Ad8, Ad9, and Ad10 can combined into a separate group. They are characterized by the most branched structures of substituent R1 (for all these adducts, the first carbon atom of R1 is the CH-group with two alkyl substituents, and thus R1 can be written with the general formula CH(X)–X′). In this group, the adducts are ranked Ad5, Ad8, Ad10, and Ad9, following the order of the ease of reactivation; moreover, these adducts are the most difficult to reactivate among all those studied (the last four rows in Table 6). In this group, the complexity of reactivation increases with increasing branching of substituent R1. The most easily reactivated adduct in this group is Ad5. In Ad5, the distance between the phosphorus and fluorine atoms is shorter than in Ad8, Ad9, and Ad10, which leads to a more negative value of E_el_P–F and a lower energy barrier. According to our calculations, Ad8 is reactivated better than Ad9 and Ad10. In Ad8, substituent R1 is a massive hexane ring, but in the active site of BChE, it can be situated in an extended conformation (Figure 11H), leaving space for the fluoride ion to approach the phosphorus atom.

For adducts Ad9 and Ad10 (as stereoisomers at the first carbon atom), the values of ΔE_el_ are also different. The data analysis in Table 5 showed that this difference is due mainly to different charges on atoms P and Oγ of these adducts. In adduct Ad9, the massive tert-butyl group of substituent R1 is directed on the opposite side from the methyl group of substituent R2, which gives a place for the fourth water molecule in the active site (Figure 11I). This water molecule forms a hydrogen bond with the oxygen atom of the O-alkyl moiety of the adduct and affects the distribution of the charges.

Adducts Ad4, Ad6, and Ad7 can be placed in another separate group with the substituent R1 described by the general formula O–CH_2_–Y. In this group, the adducts are ranked Ad7, Ad6, and Ad4 according to the degree of difficulty of reactivation. The most easily reactivated adduct in this group is Ad7. If we compare its energy and geometric characteristics with Ad6, we can see that the charge distribution of Ad5 and Ad7 is more similar to each other than to Ad4. However, because of the lower branching of substituent R1 in Ad7, the fluorine and phosphorus atoms can approach each other at a shorter distance compared with Ad6, which reduces the value of E_el_P–F and lowers the energy barrier of the reaction. When comparing Ad7 and Ad4, it is noted that the values of distP–F for these adducts are close to each other (since substituent R1 is linear in both adducts). However, the length of the substituent has a significant effect on the charge distribution in the system. As a result, the energy characteristics are different for these adducts. Ad4 and Ad6 are reactivated with similar efficiency (Ad6 is slightly more efficient). However, this result is achieved by two opposing factors. In the case of branched Ad6, the fluoride ion is located farther from the phosphorus atom of the adduct, resulting in a higher value of E_el_P–F for Ad6 than for Ad4. However, the energy barrier for Ad6 is compensated by a higher value of E_el_P–Oγ caused by a different charge distribution.

Adduct Ad3 can be distinguished separately from the groups of adducts described above. As noted in the previous section, its R2 substituent is more massive than in Ad4 but less massive than in Ad1 and Ad2 (Table 6). Therefore, in the case of Ad3, the third water molecule can be placed in the active center of BChE, but because of the ethyl substituent R2 (more massive compared with Ad4 and more hydrophobic compared with Ad1 and Ad2), the hydrated fluoride ion moves further from the phosphorus atom than in the case of Ad4 (Table 4 and Table 5). It increases the energy barrier and complicates the reactivation process.

### 2.10. Site-Selectivity of the Interaction of Fluoride Ion with HSA Adducts

For Ad12, we evaluated the efficiency of OP release from adducts with different sites of HSA. First, we performed the conformational analysis of adducts Ad12-Tyr150, Ad12-Tyr161, and Ad12-Tyr411 (Figure 5), determining whether there are amino acids in the environment of the adducts that can potentially stabilize the position of the fluoride ion in the sites of modification (in BChE, His438 plays the role of the stabilizer). The geometry of adduct Ad12-Tyr150 (Figure 5A) is such that one of the hydrogen atoms of the guanidine group of Arg257 can potentially stabilize the position of the fluoride ion near the phosphorus atom of the adduct. There are no amino acids in the environment of Ad12-Tyr161 that could stabilize the fluoride ion near the phosphorus atom of the adduct (Figure 5B). In Ad12-Tyr411, Lys414 can potentially stabilize the position of the fluoride ion in the productive conformation near the phosphorus atom of the adduct (Figure 5C). 

Similar to BChE (Section 2.5), we performed molecular docking of fluoride ion near HSA adducts Ad12-Tyr150 and Ad12-Tyr411, then checked the stability of the obtained complexes by MD simulation. The results of molecular docking and MD are presented in Figure 13 and Figure 14. 

In the case of adduct Ad12-Tyr150, the fluoride ion is located between one of the hydrogen atoms of the guanidine group of Arg257 and the phosphorus atom of the adduct at the starting point of the simulation (Figure 13A). However, quite soon, the system transforms to another conformation in which the position of the fluoride ion is stabilized by two hydrogen atoms of the guanidine group of Arg257 and three water molecules (Figure 13B). The fluoride ion stays relatively close to the phosphorus atom for 4 ns, then moves away along with the side chain of Arg257 (Figure 13C). As performed for BChE, we calculated the average distance between the atoms of phosphorus and fluorine over the first ns of simulation; its value was 3.80 Å (Table 7), which is even greater than for the most resistant BChE adducts (Table 4).

In the case of adduct Ad12-Tyr411, at the starting point of the simulation, the fluoride ion is located between one of the hydrogen atoms of the NH_3_-group of Lys414 and the phosphorus atom (Figure 14A) of the adduct. Almost immediately, the system transforms to another conformation, in which the position of the fluoride ion is stabilized by four water molecules and the side chains of Arg410 and Lys414; however, the fluoride ion does not interact with the organophosphorus moiety of the adduct (Figure 14B). The distance between the atoms of fluorine and phosphorus goes beyond 4 Å (distance suitable for nucleophilic attack) in the first 75 ps of the simulation (Figure 14C). The average distance between fluorine and phosphorus for the first ns of the simulation is 5.71 Å.

Using the AM1 method, we calculated the value ΔE_el_ for adduct Ad12-Tyr150 as equal to 39.698 kcal/mol (Table 7), which, according to the MM approach, is higher than that for the hardest reactivatable BChE adducts (Table 6).

Thus, according to our assessment, HSA adducts are much less efficiently cleaved by fluoride ion compared with BChE, and the probability of cleavage in the adducts with Tyr150 is higher than that those with Tyr411.

### 2.11. Stereoselectivity of the Interaction of Fluoride Ion with HSA Adducts

Using adducts Ad13 and Ad14 with Tyr150 of HSA (Ad13-Tyr150 and Ad14-Tyr150), we studied how the stereometry of the phosphorus atom affects the ability of the adducts to be cleaved by fluoride ion. Ad13 and Ad14 adducts were chosen because it was previously experimentally revealed that these adducts with HSA can be cleaved by fluoride ion [21]. Analogous to adduct Ad12-Tyr150, we performed molecular docking of fluoride ion near the adducts and then checked the stability of the obtained complexes by MD simulation. The results of molecular docking and dynamics are presented in Figure 15 and Figure 16.

In the case of the adduct Ad13-Tyr150, at the starting point of the simulation, the fluoride ion is located between one of the hydrogen atoms of the guanidine group of Arg257 and the phosphorus atom of the adduct (Figure 15A). However, the system soon transforms into another conformation, in which the position of the fluoride ion is stabilized by two hydrogen atoms of the guanidine group of Arg257 and three water molecules (Figure 15B). The fluoride ion stays relatively close to the phosphorus atom for 1 ns, then moves away along the side chain of Arg257 (Figure 15C). The average distance between the phosphorus atom and the fluoride ion over the first 1 ns of the simulation is 3.92 Å (Table 7), which is greater than that of the BChE adducts most difficult to reactivate and adduct Ad12-Tyr150 (Figure 13).

In the case of adduct Ad14-Tyr150, at the starting point of the simulation, the fluoride ion is also located between one of the hydrogen atoms of the guanidine group of Arg257 and the phosphorus atom of the adduct (Figure 16A). During 2 ns of the simulation, the system is kept in the conformation shown in Figure 16B. In this conformation, the position of the fluoride ion is stabilized by four water molecules and one of the hydrogen atoms of the guanidine group of Arg257; however, the fluoride ion is located farther from the phosphorus atom of the adduct compared with the R-isomer (Ad13-Tyr150). The average distance between the atoms of fluorine and phosphorus for the first 1 ns of the simulation is 4.88 Å (Table 7). After 2 ns of simulation, the fluoride ion leaves the site (Figure 16C).

Thus, according to our computational experiment, OP release should be more effective from Ad13 (R-stereoisomer) compared with Ad14 (S-stereoisomer). We assume that this difference is due to the fact that atom Nε of Arg257 of the S-enantiomer tends to form a hydrogen bond with the nitrogen atom of substituent R2 of the adduct (Figure 15B), displacing the fluoride ion from the space near the phosphorus atom. In the R-isomer, the O-ethyl substituent is located in place of the N-dimethyl substituent (Figure 15A). The geometry of the guanidine group of Arg257 and substituent R1 in the complex with the fluoride ion facilitates atom Nε to interact with the fluoride ion rather than with the oxygen of the O-ethyl group. Therefore, in the case of the R-isomer, the favorable position of Arg257 relative to the modified Tyr150 is different than in the S-isomer. It is expected that, in the semiempirical approach AM1, the R-isomer is also cleaved better than the S-isomer; the values of ΔE_el_ are 39.659 kcal/mol for the R-isomer (which is comparable with adduct Ad12-Tyr150) and 43.281 kcal/mol for the S-isomer (Table 7). This result suggests that the efficiency of OP release from other adducts is also influenced by stereoselectivity, which should be considered when developing an experimental technique for determining OP exposure.

In Table 7, by analogy with BChE, we gathered the geometric and energy characteristics of the complexes of the fluoride ion with modified Tyr150 of HSA.

### 2.12. Primary In Silico Evaluation of the Possibility of the Cleavage of CBDP Adducts on HSA

It is known that dealkylated organophosphorus adducts (containing negatively charged P(O)O^−^-group) are resistant to various reactivators, including fluoride ion. HSA adducts with CBDP Ad15 (*o*-cresyl phosphotyrosine) and Ad16 (ring-opened adduct on lysine) also contain the same negatively charged P(O)O^−^-group. For this reason, they cannot be cleaved by fluoride ion, unlike the nonaged adduct of CBDP with BChE (Ad11). However, in recent years, the efforts of several researchers have been focused on the development of so-called realkylators, which are compounds capable of transferring their alkyl substituent to the P(O)O^−^-group of a dealkylated organophosphorus adduct; after realkylation, modified ChE become available for reactivation again [43]. Zhuang et al. [44] demonstrated the ability of the quinone methide precursors (combining the properties of a realkylator and a reactivator) to restore in vitro the activity of AChE modified with dealkylated organophosphorus adducts. The authors also studied the interaction of the most reactive compound, C8, with the active site of modified AChE using molecular modeling methods and showed that the zwitterionic forms of compound C8 can bind in the active site of modified AChE in the productive conformation (in which the nucleophilic attack of the phosphoryl oxygen of the adduct on benzylic carbon of C8 is possible).

Analogously to [44], we performed molecular docking in the zwitterionic form of compound C8 near HSA adducts Ad15 and Ad16. The result of molecular docking is shown in Figure 17.

According to the obtained data, in site Tyr411, the distance between the functionally significant atoms of Ad15 and C8 is 4.40 Å, which is too far for the nucleophilic attack of the phosphoryl oxygen of the adduct on the benzylic carbon of C8. This distance in the Lys414 site is 3.37 Å, sufficient for the nucleophilic attack. The calculated values of the free binding energy of the C8 complexes with Ad15 and Ad16 are −3.5 kcal/mol for both adducts.

## 3. Discussion

In Table 8, we have gathered known experimental data on the efficiency of OP release from modified plasma proteins when exposed to fluoride ion and the results of our in silico studies. Other reports describe the procedure of ChE reactivation with fluoride ion [45,46,47]. However, in these studies, only one or two OPs are discussed, and none performed the comparative structural–functional analysis that would explain the different degrees of reactivation efficiency.

As follows from the values given in Table 8, different in vitro and in vivo experiments give contradicting results when ranking adducts according to the efficiency of OP release. The conditions of our in silico experiments are closest to those of in vitro experiments with purified human (hBChE) and rhesus macaque BChE (rmBChE) [17]. As noted above, BChE adducts studied in our work can be conditionally divided into three groups according to their structure. In group I, we included adducts Ad1 and Ad2 characterized by massive substituent R2 with an electronegative atom (nitrogen or oxygen). In group II, we included adducts Ad5, Ad8, Ad9, and Ad10 (their branched substituent R1 can be written with the general formula CH(X)–X’). Group III includes adducts Ad4, Ad6, and Ad7, whose R1 substituent is less branched than in group II and can be described by the general formula O–CH_2_–Y. Adduct Ad3 is transitional between group I and group III. Finally, adduct Ad11, which, due to its massive aromatic substituents R1 and R2, cannot be included in any described groups and should be considered separately. 

We now consider the consistency of experimental and calculated data for each group. For adduct Ad1, our calculation agrees well with the experimental data. This adduct is the least resistant to fluoride ion exposure in the experiments with human and macaque BChE and the computational experiment using the MM approach; according to the AM1 approach, Ad1 is the second on the list (Table 8). In the experiment with human BChE in vitro, OP release from Ad2 was at the level of Ad5, whereas according to molecular modeling data, this adduct is cleaved much easier at the level of Ad1. We believe this effect may be associated with spontaneous hydrolysis and aging of adduct Ad2, which were not taken into account in the computational experiment but occur under real conditions in vitro.

For Ad5, Ad8, and Ad9/Ad10 (the adducts whose substituent R1 can be written with the general formula O–CH(X)–X’), the obtained ranking is generally consistent with experimental data on the reactivation of hBChE and rmBChE (Table 8). OP release from adduct Ad10 is more productive than from Ad8 according to the MM approach and less productive according to the AM1 approach. Van der Schans et al. [17] failed to detect OP release from adducts Ad9/Ad10 (mixture of the isomers) in the experiment with hBChE because of the high rate of aging of this mixture. In the case of rmBChE, the mixture of Ad9/Ad10 was cleaved better than Ad8. However, both in the case of our theoretical calculations and in the experiments in vitro, the productivity of OP release from Ad8, Ad9, and Ad10 was closer to each other than to other adducts. The results of in vivo and in vitro experiments with plasma/blood are highly controversial (Table 8) because too many different factors can affect the effectiveness of OP release. However, in all experiments with plasma/blood, the Ad9/Ad10 adducts mixture shows cleavage inferior to Ad8 (Table 8), which is consistent with our computational data and inconsistent with in vitro data with rmBChE. Since the active sites of ChE are highly conserved in mammals, the ranking of these adducts is probably influenced not by interspecies differences but rather by selected conditions in various in vitro experiments and by simplifications in our computational experiments. Both the MM and AM1 methods showed that adduct Ad10 is cleaved more easily than its stereoisomer Ad9 (effect explained above in Section 2.9), but it was not possible to compare this result with the experiment because these enantiomers were not separated in the published papers on ChE reactivation by fluoride ion.

The adducts of group III (Ad4, Ad6, and Ad7), whose substituent R2 is the methyl group and substituent R1 can be written with the general formula O–CH_2_–Y, are ranked in the sequence Ad7, Ad4, and Ad6 according to the MM approach and in the sequence Ad7, Ad6, and Ad4 according to the AM1 approach. However, according to the calculated values of <distP–F> and ΔE_el_ (Table 4 and Table 6), adducts Ad4 and Ad6 are closer to each other in terms of OP release productivity than to Ad7. The simplicity of Ad7 cleavage is a result of its linear substituent R1 in an extended conformation, occupying less space inside the active center of BChE compared with more branched substituents R1 of adducts Ad4 and Ad6. In the available literature, we found no comparative analysis of OP release from Ad4, Ad6, and Ad7 in experiments in vitro with purified BChE. In the in vitro experiment by Seto et al. [16] using blood plasma, the ranking of Ad4 and Ad6 coincides with our calculated data using the AM1 approach. However, in our computational experiment, the difference in the cleavage efficiency for these two adducts is significantly lower than in the experiment described in [16]. In the experiment by Koller et al. [20] with human blood plasma, OP release from Ad7 was less productive compared with Ad4, which does not agree with our result. As noted above, different results may be associated with different degrees of spontaneous reactivation of modified BChE depending on adduct structure and different experimental conditions. In the experiment with purified BChE [17], only adduct Ad4 of group III was studied. Because there are no experimental data to compare the cleavage ability between the group III adducts, we compared the results for Ad4 to those from other groups. According to both the mM and AM1 approaches, Ad4 is cleaved slightly better than Ad5, which is inconsistent with in vitro-derived data with purified BChE (for both human and macaque BChE, Ad5 showed better cleavage than Ad4). This inconsistency may be due to Ad4 being more susceptible to spontaneous reactivation [17] than other adducts. For this reason, the yield of *O*-ethylmethylfluorophosphonate (the product of Ad4 interaction with fluoride ion) observed in the experiment is lower than it could be in the absence of spontaneous reactivation; hence the distortion in the productivity of fluoroanhydride release from Ad4 when exposed to fluoride ion. Since spontaneous reactivation is not taken into account in our theoretical calculations, there is a discrepancy between theoretical and experimental data.

In the case of adduct Ad3, we did not find published experimental data on the efficiency of BChE reactivation. According to our calculations, the distance between the phosphorus atom of adduct Ad3 and the fluoride ion is larger compared with adduct Ad4 with a similar structure (substituents R2 of Ad3 and Ad4 are ethyl and methyl groups, respectively) because the more massive hydrophobic ethyl group of adduct Ad3 repulses the fluoride ion hydrated with three water molecules. Because of this, Ad3 should be cleaved rather weakly. Similarly, for adduct Ad11 (CBDP adduct), we did not find published experimental data on the efficiency of its cleavage with fluoride ion. According to our calculations, because of the intramolecular hydrogen bond, the geometry of the active site of BChE inhibited by CBDP is similar to the geometry of adduct Ad2, which allows adduct Ad11 to be easily cleaved by fluoride ion.

Thus, according to the obtained data, in the case of the branched adducts with the general formula of the *O*-alkyl substituent O–CH(X)–X′, BChE reactivation efficiency decreases with the increasing steric size of the group X′. The charge distribution plays a less important role so that the reactivation efficiency can be predicted even with the MM approach. In the case of the adducts with the general formula of the *O*-alkyl substituent O–CH_2_–Y, preliminary analysis of the reactivation efficiency requires at least a semiempirical approach, since for the adducts of this group—due to the lower branching of the alkyl substituent—the influence of the steric dimensions of group Y is weaker compared with group (X)–X′, and the distribution of charges on functionally significant atoms has an additional effect on the configuration of the productive complex.

It is interesting to compare our results and research data on the reactivation of ChE with KF with the known data on the reactivation of ChE with oximes. Unlike toxic KF, which is used for analytical purposes only, oximes are primarily developed for antidote therapy, so the requirements for the effectiveness of KF and oximes are different. The effectiveness of KF is assessed mainly by the yield of OPs or their fluoroanhydrides. Oxime effectiveness can be quantified in vitro by the determination of the reactivity (k_r_) and affinity constants (1/K_D_) [48]. In a number of studies, a certain trend in the relationship between the structure of the adduct and the efficiency of reactivation was revealed for some oximes. Horn et al. [49] studied the reactivation of human BChE by bispyridinium oximes. Three adducts, Ad1, Ad2, and Ad8, were studied. For most of the oximes studied, adduct Ad1 was the slowest to reactivate, and adduct Ad8 was the fastest, which means that the rate of reactivation increased with increasing R1 substituent and decreased with increasing R2 substituent of the adducts. Luo et al. [50] revealed a similar tendency. The authors studied the reactivation of AChE from different species (rhesus monkey, human, and guinea pig) by H-oximes. Results demonstrate that for nerve agent-inhibited rhesus monkeys and human AChEs, reactivation by H-oximes accelerated as the size of the R1 substitution was increased. However, accumulated data generally indicates that the efficiency of reactivation by oximes depends not only on the structure of the adduct but also on the structure of the oxime itself. Thus, Worek et al. [51] investigated the reactivation of AChE of different species, including human AChE, by well-known oximes (obidoxime, 2-PAM, HI-6, HLö-7). For human AChE, obidoxime and 2-PAM were maximally effective in reactivating adducts Ad4 and Ad5, while HI-6 and HLö-7, in contrast, were maximally effective in reactivating Ad8. Later, Worek et al. [52] investigated the reactivation of human AChE by bispyridinium oximes by studying adducts Ad1, Ad2, and Ad8. The authors showed that the position of the oxime group(s) is decisive for the reactivating potency and that different positions of the oxime groups are important for different OP inhibitors, underlining the difficulty of developing a broad-spectrum oxime reactivator efficient against structurally different OP inhibitors.

In the case of experiments with blood plasma or animal experiments in vivo, it is important to consider that when exposed to KF, OPs could be released not only from BChE adducts but also from adducts with other proteins, in particular with albumin [21]. Here, we have shown that, first, adducts with albumin are much less efficiently cleaved by fluoride ion compared with BChE and, second, that the adducts with Tyr150 (which is less reactive when interacting with OPs compared with Tyr411) are cleaved more efficiently than adducts with Tyr411.

Regarding the interaction of HSA with CDBP, it remains unclear which amino acid CBDP would form adducts at nonsaturating concentrations, His242 or Tyr411. According to molecular modeling data, the interaction with His242 in Sudlow site I is preferable, but according to many experimental studies, Tyr411 is the most reactive amino acid in the interaction of albumin with OPs. A possible explanation might involve the formation of histidine adducts with CBDP during which, according to [31], atom Nδ of in the imidazole ring is protonated, while atom Nε is deprotonated (hid-form). From these data, we manually set up the hid-form of His242 in our HSA model. However, according to the GROMACS package (University of Groningen, the Netherlands) [53] that calculates the protonation state of amino acids depending on the local environment, His242 is in the hie-form (atom Nδ of the imidazole ring is deprotonated, and atom Nε is protonated) in free HSA. This means the equilibrium between the hie and hid tautomers is shifted towards the hie-form. Thus, considering that the hid-form of His242 is not preferred, it is likely that the interaction of low concentrations of CBDP with albumin will form an adduct with Tyr411, which agrees with the experimental data. The protonation state of the reactive residues, depending on their nearest amino acid environment, should also be taken into account when evaluating the efficiency of adduct formation and OP release after reactivator exposure.

Thus, according to our data, the proportion of fluoridates released from albumin is less than that released from the active site of BChE. Nevertheless, spontaneous dephosphorylation of HSA adducts cannot be ignored, which can lead to discrepancies between calculated and experimental data. The Spanish scientists led by E. Vilanova [26,54,55] performed in vivo experiments and found that the level of detoxification of toxicologically relevant concentrations of paraoxon with albumin was no less than detoxification with paraoxonase (PON1) so that sensitivity to paraoxon in PON1 knockout mice did not differ from the control group [26]. The authors attribute the main role in the detoxification of OPs with albumin to catalytic hydrolysis, the second stage of which is the dephosphorylation of the adduct, or spontaneous reactivation. Using nuclear magnetic resonance (NMR), we studied the hydrolysis of the nontoxic analog of paraoxon *p*-nitrophenyl acetate (NPA) in the presence of BSA and showed that acetate, which is the product of the complete hydrolysis of NPA, appears in the reaction mixture already in the first minutes of the reaction [56]. The hydrolysis of paraoxon itself is much slower [27]; nevertheless, the true esterase activity of albumin towards OPs cannot be ignored. According to our early assumptions, when esters and OPs interact with albumin, the lifetime of adducts with Tyr150 is much shorter compared with Tyr411, i.e., Tyr150 can be considered the site of true esterase activity, and Tyr411 as the site of pseudoesterase activity of albumin [28,57]. The result of the presented work indicates that modified Tyr150 is easier to reactivate with fluoride ion compared with Tyr411, which, at least, does not contradict our assumptions.

As for OP release from dealkylated CBDP adducts on HSA, according to our calculations, the possibility of modified Lys414 realkylation (and further reactivation) with quinone methide precursors is not ruled out. However, only Lys414, which is not the most reactive residue in HSA, can undergo realkylation with C8 (one of the most reactive AChE realkylators [44]). Furthermore, the value of free binding energy of the complex of C8 with modified Lys414 is relatively low (−3.5 kcal/mol), which means a meager yield of the realkylation product even at very high concentrations of C8. The set of effective realkylators for HSA will differ from that for ChE. They have yet to be found (including by molecular modeling methods), and consequently, the feasibility of using realkylators for HSA has to be evaluated by already existing methods of searching for adducts.

The practical value of the proposed approach is that it offers new possibilities to predict whether or not it is reasonable to include the result of reactivation of plasma proteins modified by a specific OP in the list of biomarkers for molecular verification of exposure to OPs. The number of known OPs and the variety of structural variations of alkyl substituents is enormous. When developing an algorithm for applying procedures for establishing the fact of OP exposure, it is important to assess the chances of detectable OP release from BChE and HSA adducts based on their structure since this procedure consumes a fairly large volume of blood plasma, whereas the amount of biological sample available for analysis is usually limited.

## 4. Limitations

Undoubtedly, our approach proposed for evaluating the effectiveness of plasma protein reactivation is not without limitations and can be improved in the future. In order to simplify the evaluation procedure and reduce the computational time, we estimated the efficiency of fluoride ion exposure only by the distance between the fluorine and phosphorus atoms of the adduct when using the MM approach and took into account only electrostatic forces (which are dominant in the fluorine–adduct interaction) when using the AM1 approach. Our method does not consider the possibility of spontaneous reactivation or dealkylation (aging), and it also does not take into account a different efficiency for OPs in forming adducts with BChE and HSA. For example, Mangas et al. [58] state that resistance to reactivation is at least partially explained by two possible mechanisms of increased adduct stability after aging and the ability of aged ChE to electrostatically repulse the negatively charged reactivators. The mechanisms of ChE aging are described in detail in the review of Masson et al. [59]. Briefly, the rate of aging depends on ChE (in general, BChE ages faster than AChE), temperature, pH, allosteric modulation of ChE 3d-structure via binding of the ligands in peripheral anionic site, and the OP structure (branched alkyl moieties dealkylate much faster than short ones). Thus, the half-time of aging ranges from a few minutes (for Ad9/Ad10) to several days (for Ad4) [59], so the yield of OPs or their fluoroanhydrides may be significantly impaired in fast aging. Further calculations taking into account all these factors will help to improve the accuracy of the computer assessment of the effectiveness of OP release from HSA and BChE adducts when exposed to fluoride ion or other reactivators.

## 5. Methods

### 5.1. Preparation of the Three-Dimensional Models of the Adducts

Three-dimensional models of low-molecular ligands were built using the HyperChem 8.0.8 program (Hypercube Inc., Gainesville, FL, USA) [60]. The three-dimensional structures of proteins were picked up from the PDB database (accessed on 1 September 2023) [32]. The following PDB entries were used: 3djy [33] and 2xqk [34] for BChE and 2bxg [36] for HSA. The models of the adduct were built as follows: The topologies of modified amino acids were described using the available information on the atomic charges, bond lengths, bond and torsion angles for different types of atoms, and atomic groups presented in the database force field parameters files of the GROMACS 2019.4 software package [53]. The prepared topologies were added to the library of GROMACS software 2019.4 [53]. Then, molecular docking of organophosphate moieties into BChE and HSA modification sites was performed (the docking procedure is described in Section 5.2). The models of modified amino acids were based on the topologies added into the library and generated from the resulting complexes using GROMACS 2019.4 software [53]. In the case of BChE, the obtained structures corresponded to the known experimental data on the conformation of organophosphorus adducts (the phosphoryl oxygen is bound at the oxyanion center, and His438 interacts with the oxygen of the *O*-alkyl substituent), so the models obtained were optimized only by the energy minimization method. For HSA, in the absence of experimental data for comparison, the structures of the adducts were optimized by 50 ns MD simulation (details of MD are described in Section 5.3). The conformations of BChE adducts after energy minimization and HSA adducts after 50 ns of simulation were used for further docking of the fluoride ion. Histidines at the modification sites were set to be double protonated since the reactivation with KF in vitro was performed in an acidic medium.

### 5.2. Molecular Docking

Molecular docking of organophosphate moieties into BChE and HSA sites of modification, as well as docking of CBDP and compound C8 into HSA sites of modification, was performed using Autodock Vina 1.1.2 software (The Scripps Research Institute, La Jolla, CA, USA) [61]. A search area of 15 × 15 × 15 Å^3^ was set in the studied protein binding site. The parameter “exhaustiveness” (determining the number of runs and the amount of computational effort) was set to 20. The parameter “energy_range” (maximum energy difference between the best and the worst binding modes) was set to 3 kcal/mol. The number of the most optimal conformations in the output file (num_modes) was set to 10. The conformation of the ligands could vary, but the protein remained rigid. For the docking of organophosphate moieties, conformations with the minimal distance between the functionally significant atoms (the phosphorus atom of organophosphate moiety and serine/tyrosine oxygen atom or histidine/lysine nitrogen atom of BChE/HSA modification sites) were selected for further optimization. In the case of docking of CBDP and compound C8, the productive conformations were selected for further analysis, namely those in which the distance between the functionally significant atoms (oxygen/nitrogen atom of the catalytically active amino acids and phosphorus/carbon atom of the ligands) does not exceed 4 Å. 

Docking the fluoride ion to modified BChE and HSA sites was performed manually. The fluoride ion was placed at a distance of 2 Å from the phosphorus atom so that the fluorine atom and bond P=O formed one straight line. The procedure consisted of calculating the coordinates of this point, which were then added via a text editor to the PDB files of the modified BChE and HSA models. The resulting structures were optimized using the energy minimization method, and the conformations of the complexes after minimization were used for further MD simulation.

### 5.3. Molecular Dynamics

The conformational changes in the adducts and the modified BChE and HSA complexes with the fluoride ion and HSA with CBDP were calculated by MD simulation with the help of GROMACS 2019.4 software [53] using force fields OPLS-AA (all-atom optimized potentials for liquid simulations [62] and CHARMM27 [63]). Each adduct/complex was virtually placed in a cubic periodic box filled with water molecules. The TIP3P [64] and TIP4P [65] water models (transferable intermolecular potential with 3 or 4 points) were used to describe water molecules. To neutralize the systems, sodium or chloride ions were added. Temperature (300 K) and pressure (1 bar) were kept constant using the V-rescale thermostat [66] and Parrinello–Rahman barostat [67]. Long-range electrostatic interactions were treated by the particle mesh Ewald method [68]. Lennard-Jones interactions were calculated with a cutoff of 1.0 nm. The LINCS algorithm (linear constraint solver for molecular simulations) was used to constrain bond length [69]. Before running the MD simulations, all the structures were minimized by the steepest descent energy minimization and equilibrated under NVT and NPT ensembles. The total length of equilibration was 1 ns for the complexes of modified BChE and HSA with fluoride ion and 6 ns for the complexes of CBDP with HSA and organophosphorus adducts on HSA. The simulation length was chosen with respect to a specific task: 100 ns with the integration step of 0.002 ps for the complexes of CBDP with HSA, 50 ns with the integration step of 0.002 ps for OP adducts with HSA, and 10 ns with the integration step of 0.001 ps for the complexes of modified BChE/HSA with fluoride ion.

### 5.4. Calculation of Energy and Geometric Characteristics of Adducts Using Semiempirical Method AM1

For all studied complexes of modified BChE and HSA with fluoride ion, we extracted the coordinates of the systems from the obtained molecular dynamics trajectories after 1 ns of the simulation (in the case of Ad7, after 8.5 ns; in the case of Ad13 and Ad14, after 0.5 ns). The coordinates were saved in PDB format. The obtained structures were optimized by the energy minimization method using GROMACS 2019.4 software [53]. Using the obtained conformations, we extracted the coordinates of the fluoride ion, Ser198, His438, and the water molecules closest to the fluorine ion (in the case of HSA, these were the coordinates of the fluorine ion, Tyr150, His242, and Arg257 closest to the fluorine ion water molecules). To maintain a neutral charge, two oxygen atoms and three hydrogen atoms were added to the C- and N-termini of the amino acids (Ser198 and His438 in the case of BChE; Tyr150, His242, and Arg257 in the case of HSA) using Visual Molecular Dynamics v.1.9.4a53 program (VMD, University of Illinois Urbana-Champaign, USA) [35] so that the amino acids were zwitterions. Next, using the HyperChem 8.0.8 software package [60], we again optimized the resulting systems by energy minimization and then calculated the charges of the phosphorus atom of the adducts (q_P_), the oxygen atom of OH-group of serine/tyrosine (q_Oγ/η_), and the fluorine ion (q_F_), as well as the distances between them using the AM1 approach [41]. The obtained values of charges and distances were used to calculate the energy of electrostatic interactions between these atoms (E_el_P–O and E_el_P–F) based on Coulomb’s law.

## Figures and Tables

**Figure 1 ijms-24-14819-f001:**
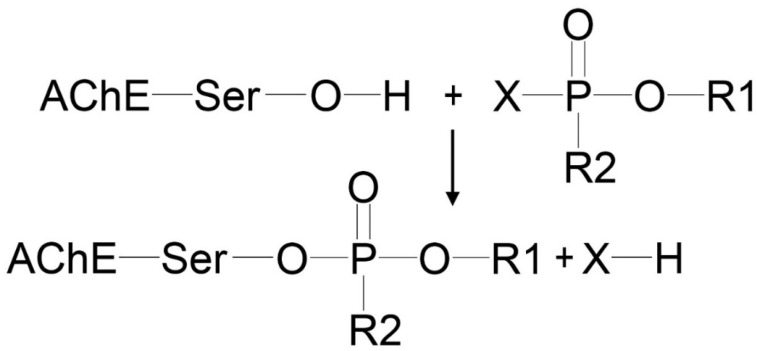
General scheme of irreversible inhibition of acetylcholinesterase (AChE) by organophosphates (OPs).

**Figure 2 ijms-24-14819-f002:**
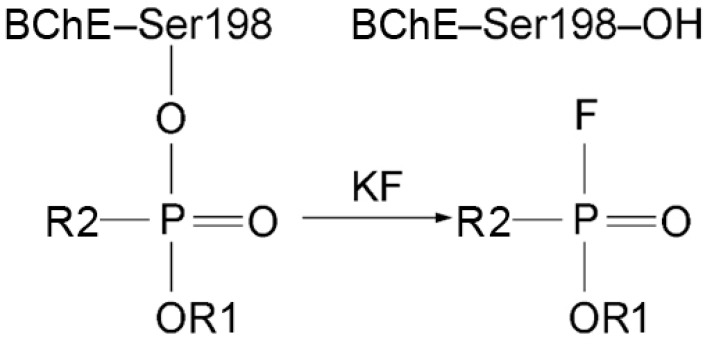
Reactivation scheme of the butyrylcholinesterase (BChE) with fluoride ion.

**Figure 3 ijms-24-14819-f003:**
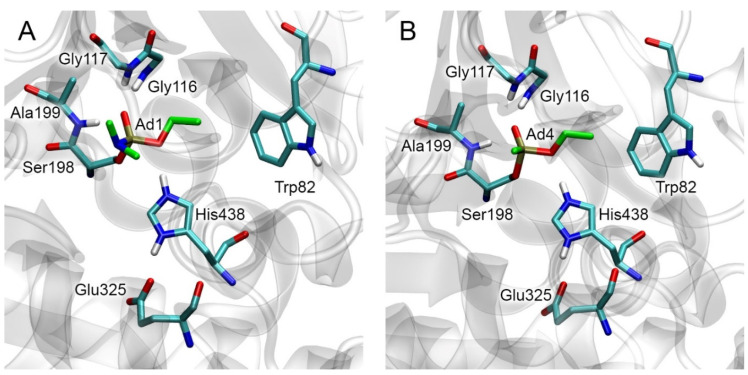
Structures of adducts Ad1 ((**A**), protein databank (PDB) entry 3djy [33], accessed on 1 September 2023) and Ad4 ((**B**), PDB entry 2xqk [34], accessed on 1 September 2023) according to X-ray analysis. Ser198, His438, and Glu325 form the catalytic triad. Trp82 is a part of the anionic center in which the ammonium cationic groups of substrates and inhibitors bind. Gly116, Gly117, and Ala199 form the oxyanion center, which is the binding site for the carboxylic oxygen of substrates and the phosphoryl oxygen of OPs. Hydrogen atoms (not resolved by X-ray) were completed by us with the help of the Visual Molecular Dynamics v.1.9.4a53 program (VMD, University of Illinois Urbana-Champaign, USA) [35]. His438 was assigned as double protonated. Carbon atoms of the organophosphorus moiety are highlighted in green. Nonessential hydrogens are omitted for clarity.

**Figure 4 ijms-24-14819-f004:**
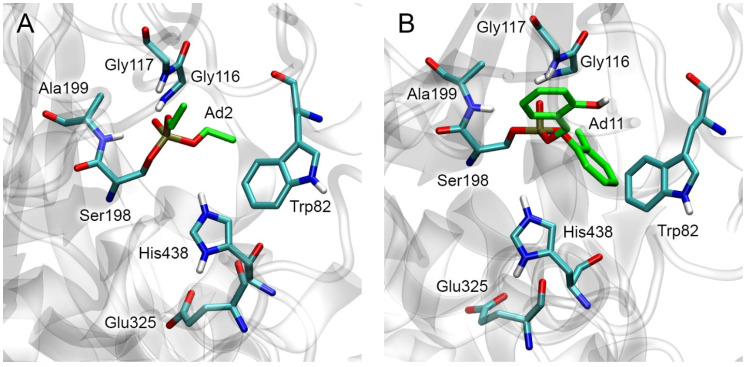
Structures of adducts Ad2 (**A**) and Ad11 (**B**) according to molecular modeling data. Carbon atoms of the organophosphorus moiety are highlighted in green. Nonessential hydrogens are omitted for clarity.

**Figure 5 ijms-24-14819-f005:**
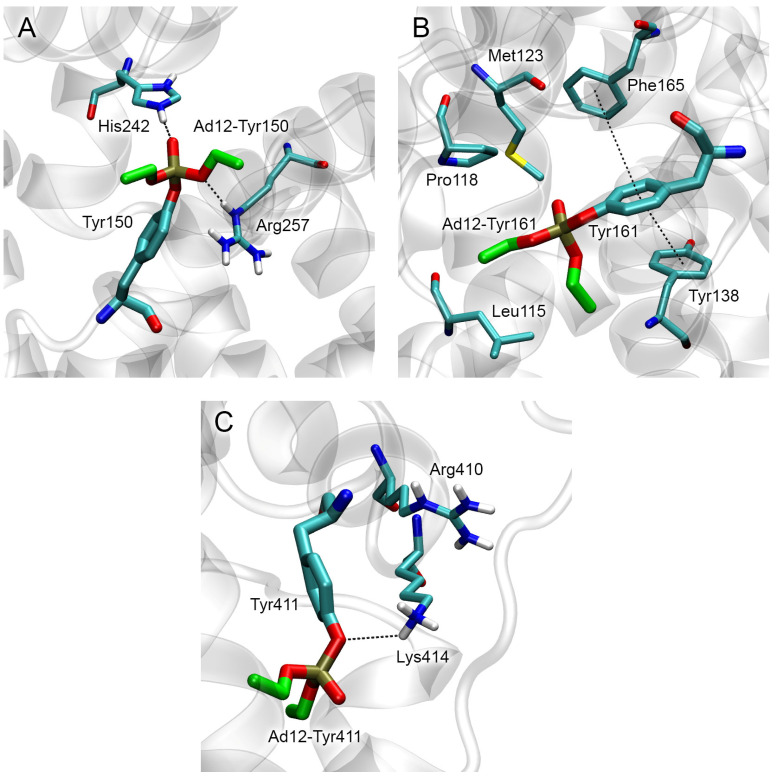
Three-dimensional models of Ad12-HSA adducts obtained by molecular modeling methods. (**A**), adduct with Tyr150 (Ad12-Tyr150); (**B**), adduct with Tyr161 (Ad12-Tyr161); (**C**), adduct with Tyr411 (Ad12-Tyr411). Carbon atoms of the organophosphorus moiety are highlighted in green. Nonessential hydrogens are omitted for clarity. Key interactions are shown by dotted lines.

**Figure 6 ijms-24-14819-f006:**
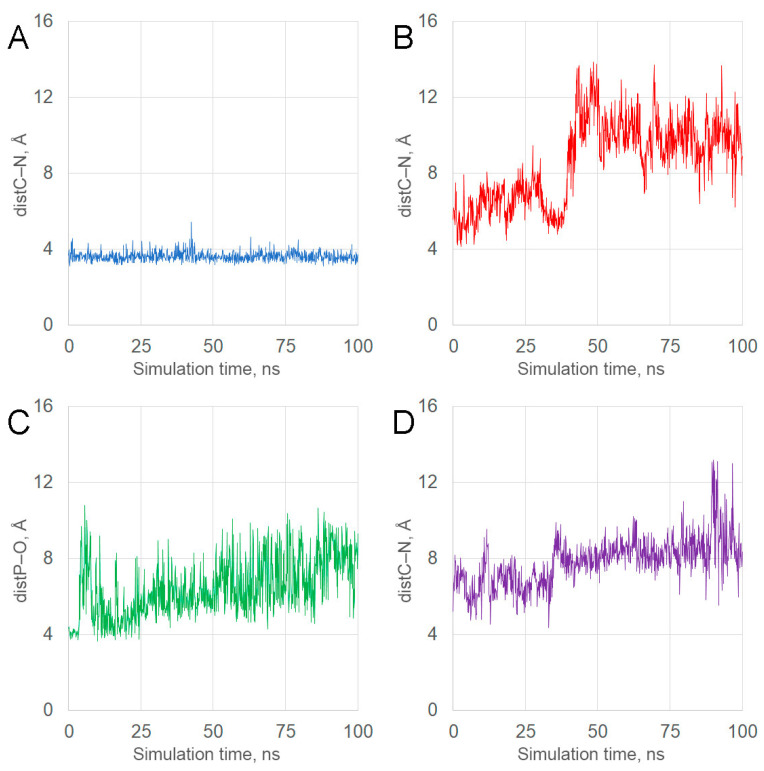
Time dependence of the distance between the functionally significant atoms in the CBDP complexes with HSA according to MD simulation. (**A**) distance between the carbon atom of the methylene group of CBDP and atom Nε of His242 (distC–N); (**B**) distance between the carbon atom of the methylene group of CBDP and atom Nζ of Lys351 (distC–N); (**C**) distance between the phosphorus atom of CBDP and the oxygen atom of the hydroxyl group of Tyr411 (distP–O); (**D**) distance between the carbon atom of the methylene group of CBDP and atom Nζ of Lys414 (distC–N).

**Figure 7 ijms-24-14819-f007:**
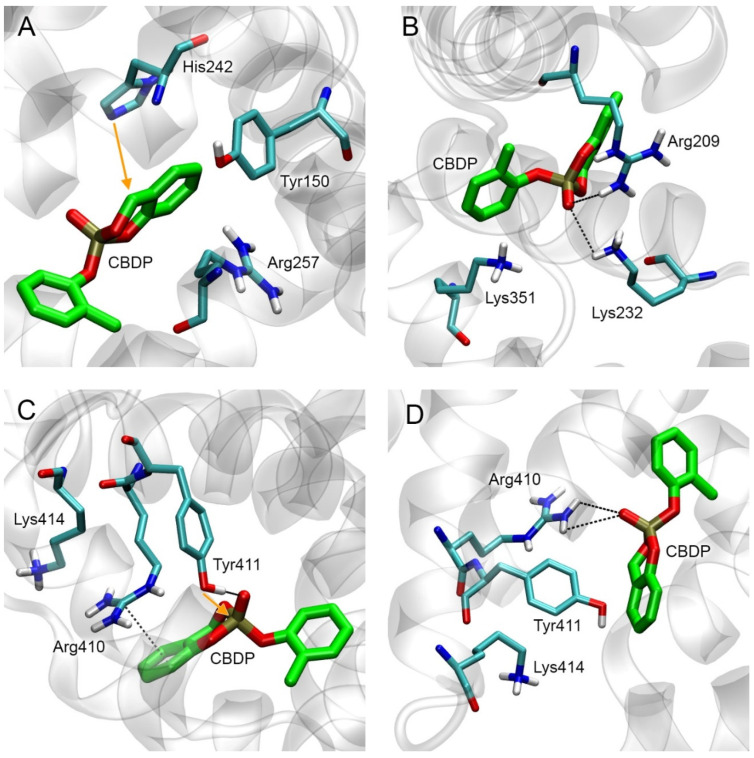
Stable conformations of CBDP at sites His242 (**A**), Lys351 (**B**), Tyr411 (**C**), and Lys414 (**D**) of HSA. The orange arrows show the direction of possible nucleophilic attack of catalytic amino acids on CBDP. Carbon atoms of CBDP are highlighted in green. Key interactions are shown by dotted lines. Nonessential hydrogens are omitted for clarity.

**Figure 8 ijms-24-14819-f008:**
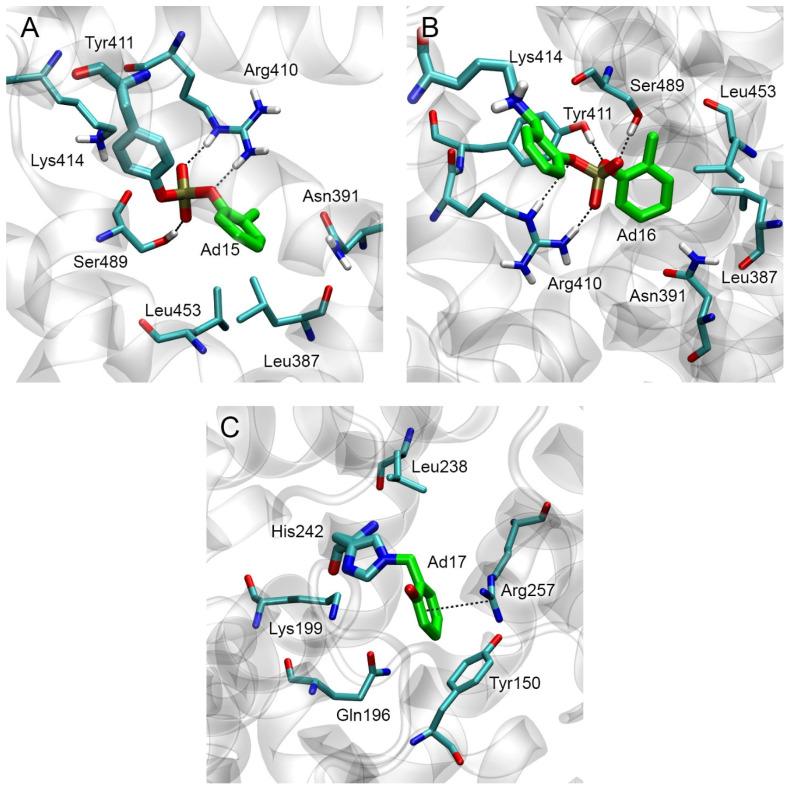
Conformations of CBDP adducts with HSA according to MD simulation. (**A**) Tyr411 adduct in site Sudlow II (Ad15); (**B**) Lys414 adduct in site Sudlow II (Ad16); (**C**) His242 adduct in site Sudlow I (Ad17). Carbon atoms of the CBDP moiety are highlighted in green. Key interactions are shown by dotted lines. Nonessential hydrogens are omitted for clarity.

**Figure 9 ijms-24-14819-f009:**
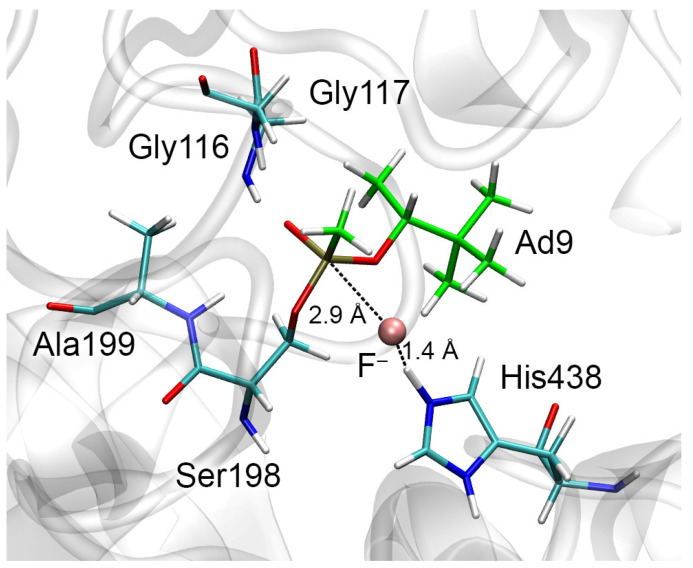
Productive complex of inhibited BChE and fluoride ion according to molecular docking on the example of adduct Ad9. The carbon atoms of the organophosphorus moiety are highlighted in green. Fluoride ion is shown as a pink sphere.

**Figure 10 ijms-24-14819-f010:**
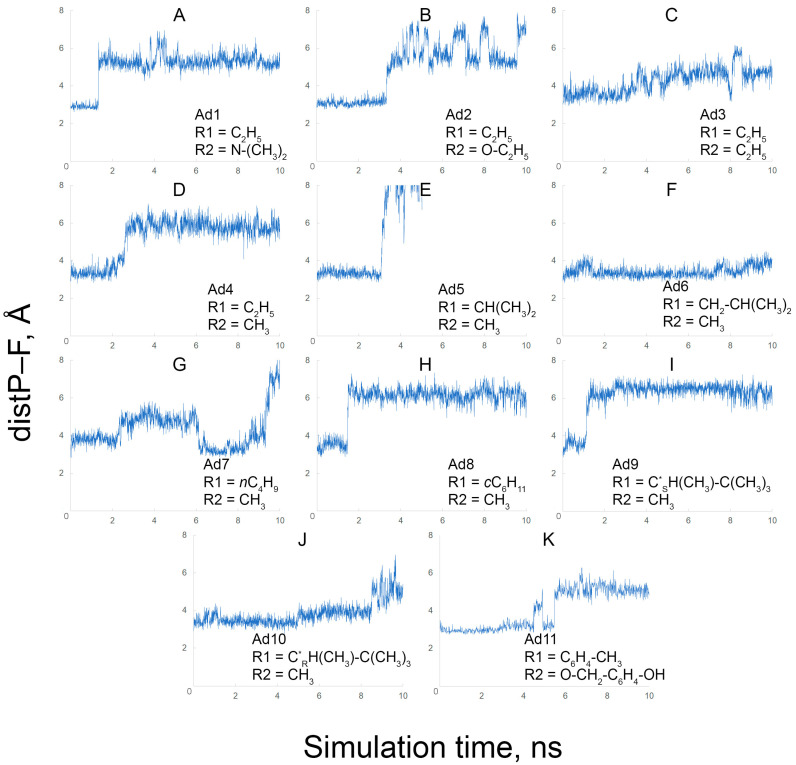
The distance between the phosphorus atom and the fluoride ion (distP–F) in BChE adducts Ad1–Ad11 (**A**–**K**) according to MD simulation. *—*R-* and *S*-stereoisomers.

**Figure 11 ijms-24-14819-f011:**
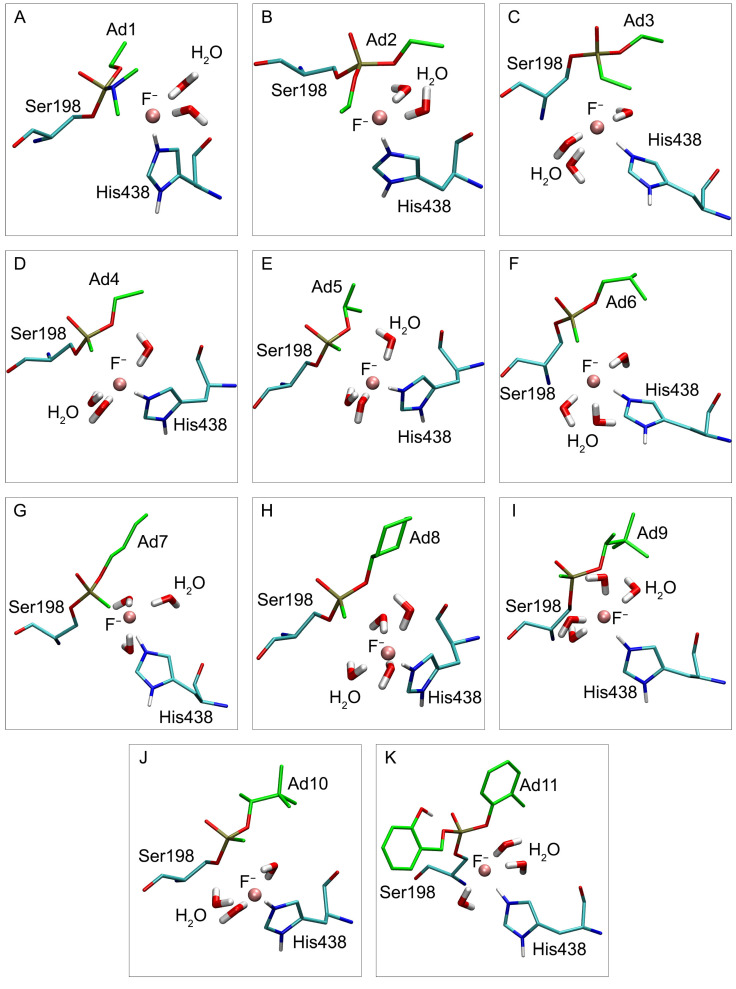
Stable, productive conformations of the fluoride ion in adducts Ad1–Ad11 (**A**–**K**) of BChE according to MD simulation. The carbon atoms of the organophosphorus moieties are highlighted in green. Fluoride ion is shown as a pink sphere. Nonessential hydrogens are omitted for clarity.

**Figure 12 ijms-24-14819-f012:**
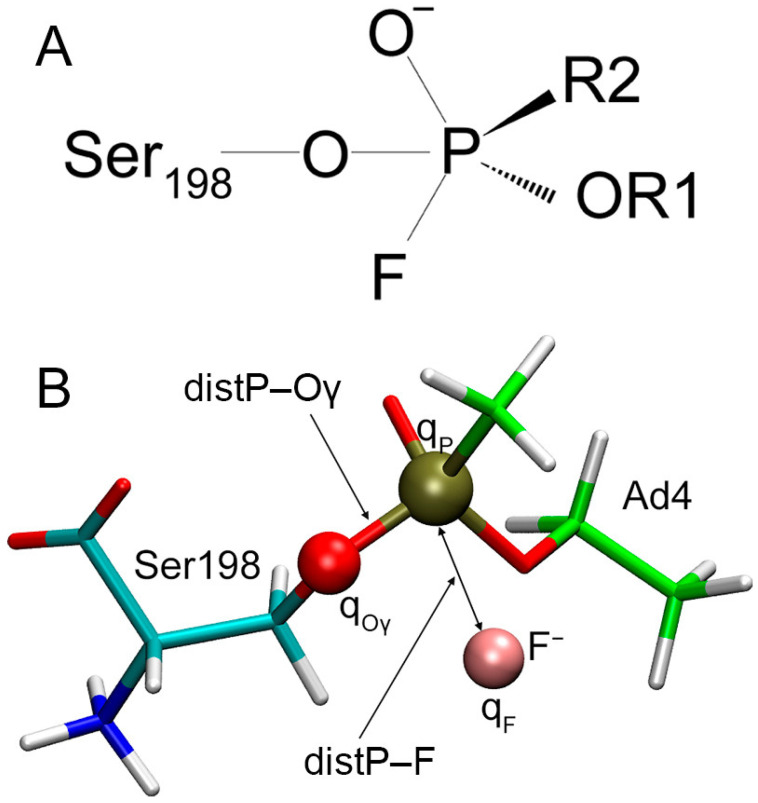
Mechanism of BChE reactivation with fluoride ion. (**A**) a proposed tetrahedral transition complex formed during the reactivation of modified BChE with fluoride ion; (**B**) legends for the charges of functionally significant atoms and the distances between them on the example of adduct Ad4 according to Austin Model 1 (AM1) approach. The carbon atoms of the organophosphorus moieties are highlighted in green. Fluoride ion is shown as a pink sphere.

**Figure 13 ijms-24-14819-f013:**
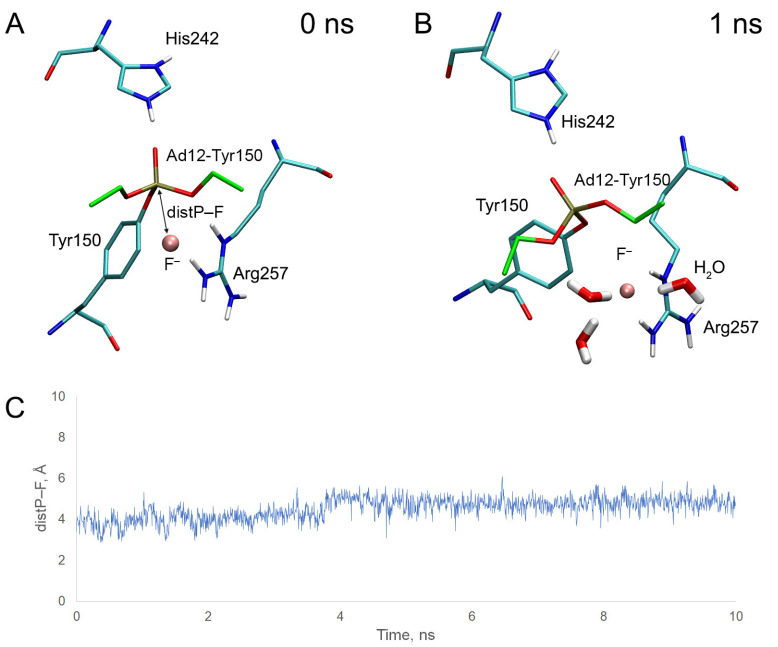
Conformation of the complex of fluoride ion with adduct Ad12-Tyr150 according to MD simulation. (**A**) starting conformation of the complex; (**B**) complex conformation after 1 ns of the simulation; (**C**) time dependence of the distance between the phosphorus atom of the adduct and the fluoride ion (distP–F). The carbon atoms of the organophosphorus moieties are highlighted in green. Fluoride ion is shown as a pink sphere. Nonessential hydrogens are omitted for clarity.

**Figure 14 ijms-24-14819-f014:**
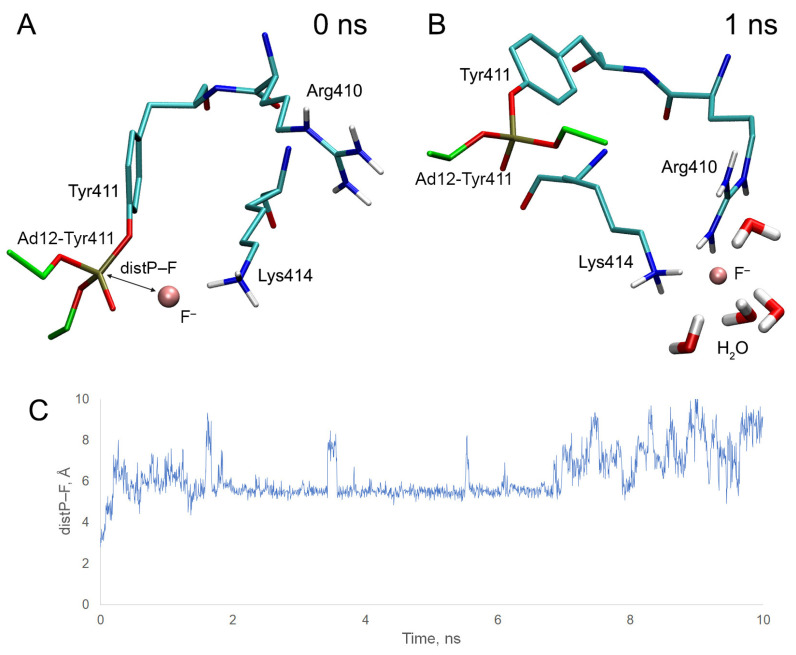
Conformation of the complex of fluoride ion with adduct Ad12-Tyr411 according to MD simulation. (**A**) starting conformation of the complex; (**B**) complex conformation after 1 ns of the simulation; (**C**) time dependence of the distance between the phosphorus atom of the adduct and the fluoride ion (distP–F). The carbon atoms of the organophosphorus moieties are highlighted in green. Fluoride ion is shown as a pink sphere. Nonessential hydrogens are omitted for clarity.

**Figure 15 ijms-24-14819-f015:**
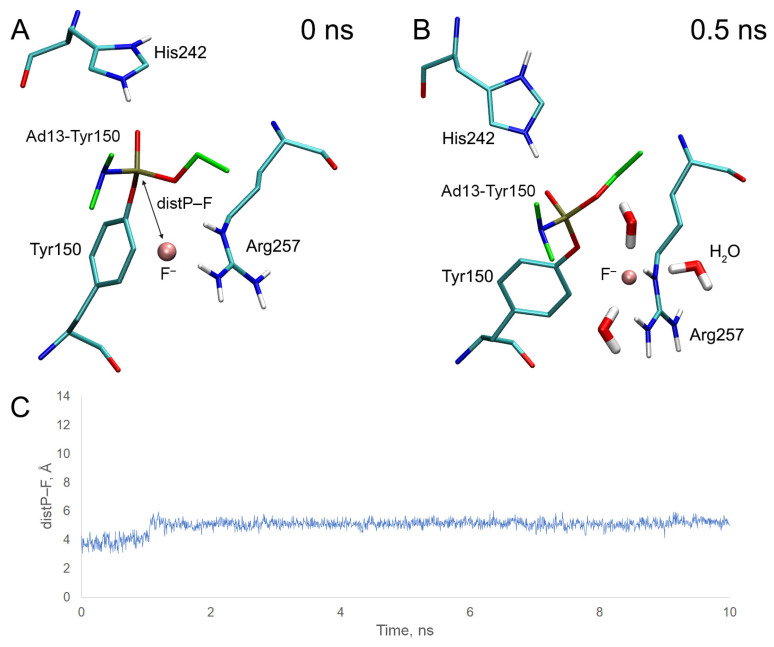
Conformation of the complex of fluoride ion with adduct Ad13-Tyr150 according to MD simulation. (**A**) starting conformation of the complex; (**B**) complex conformation after 0.5 ns (500 ps) of the simulation; (**C**) time dependence of the distance between the phosphorus atom of the adduct and the fluoride ion (distP–F). The carbon atoms of the organophosphorus moieties are highlighted in green. Fluoride ion is shown as a pink sphere. Nonessential hydrogens are omitted for clarity.

**Figure 16 ijms-24-14819-f016:**
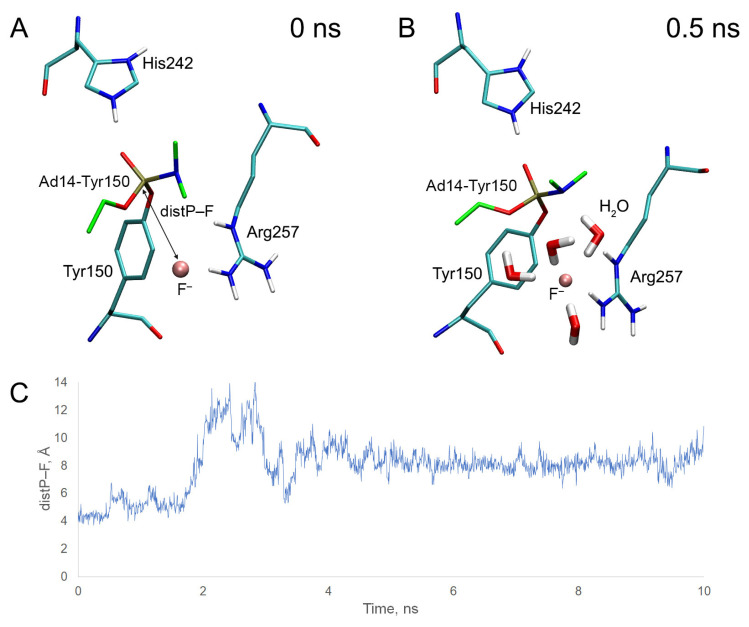
Conformation of the complex of fluoride ion with adduct Ad14-Tyr150 according to MD simulation. (**A**) starting conformation of the complex; (**B**) complex conformation after 0.5 ns (500 ps) of the simulation; (**C**) time dependence of the distance between the phosphorus atom of the adduct and the fluoride ion (distP–F). The carbon atoms of the organophosphorus moieties are highlighted in green. Fluoride ion is shown as a pink sphere. Nonessential hydrogens are omitted for clarity.

**Figure 17 ijms-24-14819-f017:**
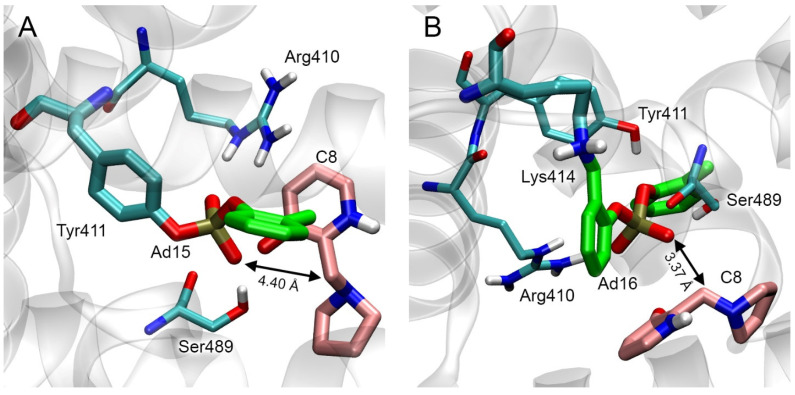
Productive conformations of realkylator C8 described in [44] in the sites of CBDP adduct formation with HSA. (**A**) adduct Ad15 with Tyr411; (**B**) adduct Ad16 with Lys414. The carbon atoms of the organophosphorus moiety are highlighted in green; the carbon atoms of compound C8 are highlighted in pink; nonessential hydrogens are omitted for clarity.

**Table 1 ijms-24-14819-t001:** General formula and structures of the studied organophosphorus adducts on butyrylcholinesterase (BChE).

Adduct	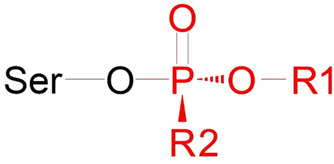	
	R1	R2
Ad1	C_2_H_5_	N-(CH_3_)_2_
Ad2	C_2_H_5_	O-C_2_H_5_
Ad3	C_2_H_5_	C_2_H_5_
Ad4	C_2_H_5_	CH_3_
Ad5	CH(CH_3_)_2_	CH_3_
Ad6	CH_2_-CH(CH_3_)_2_	CH_3_
Ad7	*n*C_4_H_9_	CH_3_
Ad8	*c*C_6_H_11_	CH_3_
Ad9	C*_S_H(CH_3_)-C(CH_3_)_3_	CH_3_
Ad10	C*_R_H(CH_3_)-C(CH_3_)_3_	CH_3_
CBDP adduct		
Ad11	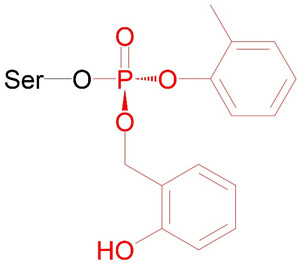	

*—*R-* and *S*-stereoisomers; CBDP—4H-1,3,2-Benzodioxaphosphorin, 2-(2-methylphenoxy)-, 2-oxide. Ser198 side chain is shown in black, organophosphorus moiety is highlighted in red.

**Table 2 ijms-24-14819-t002:** General formula and structure of the studied adducts formed during the interaction of OPs with human serum albumin (HSA).

Adduct	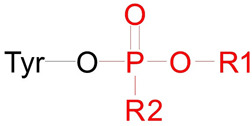	
	R1	R2
Ad12	C_2_H_5_	O-C_2_H_5_
Ad13 (P*_R_*)	C_2_H_5_	N-(CH_3_)_2_
Ad14 (P*_S_*)	C_2_H_5_	N-(CH_3_)_2_
CBDP adducts		
Ad15	*o*-cresyl phosphotyrosine	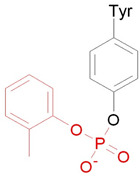
Ad16	ring-opened adduct	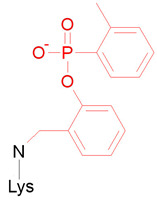
Ad17	*o*-hydroxybenzyl adduct	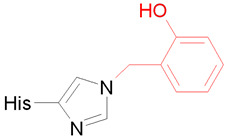

CBDP, 4H-1,3,2-Benzodioxaphosphorin, 2-(2-methylphenoxy)-, 2-oxide; HSA residues side chains are shown in black; organophosphorus fragment highlighted in red.

**Table 3 ijms-24-14819-t003:** Geometrical and energy characteristics of CBDP complexes with HSA at sites of adduct formation according to molecular docking data.

Site	distP–O/distC–N, Å	ΔG, kcal/mol
His67	3.7	−4.9
Tyr138	not found	
Tyr140	4.0	−5.4
His146	3.7	−5.6
Lys199	3.0	−5.9
His242	3.5	−7.5
His247	3.7	−4.2
His338	3.6	−5.8
Lys351	3.8	−6.9
Tyr411	3.7	−6.8
Lys414	3.3	−7.2
Lys432	not found	
Lys525	3.4	−5.3

distP–O—the distance between the hydroxyl oxygen atom of tyrosine and the phosphorus atom of CBDP; distC–N—the distance between the nitrogen atom Nε of histidine or lysine and the carbon atom of the methylene group of CBDP; ΔG—the estimated value of free binding energy; not found—productive conformations were not found.

**Table 4 ijms-24-14819-t004:** Average distances between the phosphorus atom of inhibited BChE and fluoride ion according to molecular dynamics (MD) simulation.

**Decrease in BChE Reactivation Efficiency** 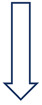	**Adduct**	**R1**	**R2**	**Hydration of F^−^,** **Water Molecules**	**<distP–F>, Å**
	Ad1	C_2_H_5_	N-(CH_3_)_2_	2	2.91
	Ad11	C_6_H_4_-CH_3_	O-CH_2_-C_6_H_4_-OH	3	2.96
	Ad2	C_2_H_5_	O-C_2_H_5_	2	3.07
	Ad7	*n*C_4_H_9_	CH_3_	3	3.18
	Ad4	C_2_H_5_	CH_3_	3	3.32
	Ad5	CH(CH_3_)_2_	CH_3_	3	3.37
	Ad6	CH_2_-CH(CH_3_)_2_	CH_3_	3	3.43
	Ad10	C*_R_H(CH_3_)-C(CH_3_)_3_	CH_3_	3	3.50
	Ad3	C_2_H_5_	C_2_H_5_	3	3.52
	Ad8	*c*C_6_H_11_	CH_3_	4	3.52
	Ad9	C*_S_H(CH_3_)-C(CH_3_)_3_	CH_3_	4	3.56

*—*R-* and *S*-stereoisomers

**Table 5 ijms-24-14819-t005:** Energy and geometric characteristics of the interaction between the functionally significant atoms in the complexes of the fluoride ion with BChE adducts according to the semiempirical method AM1.

Adduct	q_P_	q_Oγ_	q_F_	distP–Oγ, Å	distP–F, Å	E_el_P–Oγ, kcal/mol	E_el_P–F, kcal/mol	ΔE_el_, kcal/mol
Ad1	2.240	−0.618	−0.748	1.686	2.753	−68.182	−50.540	17.642
Ad2	2.377	−0.599	−0.752	1.694	2.873	−69.797	−51.666	18.131
Ad3	2.306	−0.561	−0.739	1.691	3.742	−63.529	−37.818	25.711
Ad4	2.334	−0.587	−0.726	1.688	3.073	−67.400	−45.790	21.610
Ad5	2.327	−0.602	−0.721	1.687	3.222	−68.956	−43.241	25.715
Ad6	2.284	−0.539	−0.745	1.691	3.684	−60.455	−38.355	22.100
Ad7	2.332	−0.549	−0.748	1.688	2.978	−62.983	−48.641	14.342
Ad8	2.362	−0.623	−0.703	1.692	3.763	−72.221	−36.643	35.577
Ad9	2.337	−0.647	−0.712	1.680	3.718	−74.739	−37.164	37.575
Ad10	2.299	−0.624	−0.701	1.687	3.918	−70.616	−34.158	36.458
Ad11	2.406	−0.594	−0.753	1.684	2.823	−70.475	−53.293	17.181

q_P_, q_Oγ_, q_F_—charges on atoms P, Oγ, F; distP–Oγ—P–Oγ bond length; distP–F—distance between atoms P and F; E_el_P–Oγ and E_el_P–F—energies of electrostatic interactions between pairs of atoms P–Oγ and P–F; ΔE_el_—difference between E_el_P–Oγ and E_el_P–F.

**Table 6 ijms-24-14819-t006:** Ranking of BChE adducts according to the complexity of reactivation obtained by semiempirical method AM1.

**Decrease in BChE Reactivation Efficiency** 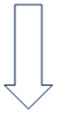	**Adduct**	**R1**	**R2**	**ΔE_el_, kcal/mol**
	Ad7	*n*C_4_H_9_	CH_3_	14.342
	Ad11	C_6_H_4_-CH_3_	O-CH_2_-C_6_H_4_-OH	17.181
	Ad1	C_2_H_5_	N-(CH_3_)_2_	17.642
	Ad2	C_2_H_5_	O-C_2_H_5_	18.131
	Ad6	CH_2_-CH(CH_3_)_2_	CH_3_	21.610
	Ad4	C_2_H_5_	CH_3_	22.100
	Ad3	C_2_H_5_	C_2_H_5_	25.711
	Ad5	CH(CH_3_)_2_	CH_3_	25.715
	Ad8	*c*C_6_H_11_	CH_3_	35.577
	Ad10	C*_R_H(CH_3_)-C(CH_3_)_3_	CH_3_	36.458
	Ad9	C*_S_H(CH_3_)-C(CH_3_)_3_	CH_3_	37.575

*—*R-* and *S*-stereoisomers

**Table 7 ijms-24-14819-t007:** Energy and geometric characteristics of the complexes of fluoride ion and modified Tyr150 of HSA according to MD simulation and semiempirical method AM1.

Adduct	Hydration of F, Water Molecules (MD)	<distP–F>, Å (MD)	ΔE_el_, kcal/mol (AM1)
Ad12-Tyr150	3	3.805	39.698
Ad13-Tyr150	3	3.920	39.659
Ad14-Tyr150	4	4.879	43.281

**Table 8 ijms-24-14819-t008:** Efficiency of OP release when exposed to fluoride ion according to in vitro, in vivo, and in silico data.

	In Vitro, hBChE [17]	In Vitro, rmBChE [17]	In Vivo, Female Macaque [17]	In Vivo, Male Macaque [17]	In Vitro, Human Serum [16]	In Vitro, Human Serum [18]	In Vitro, Human Blood [20]	In Silico (MM),hBChE or HSA	In Silico (AM1)hBChE or HSA
**Decrease in OP Release Efficiency** ^†^ **  **	Ad1	Ad1	Ad8	Ad5	Ad6	Ad8	Ad1	Ad1 ^a^	Ad7 ^a^
Ad5	Ad5	Ad5	Ad1/Ad8 *	Ad1	Ad9/Ad10	Ad5	Ad11 ^a^	Ad11 ^a^
Ad2	Ad4	Ad1	Ad4	Ad4	Ad1	Ad4	Ad2 ^a^	Ad1 ^a^
Ad8	Ad9–Ad10 #	Ad4	Ad9/Ad10 #	Ad8	Ad5	Ad8	Ad7 ^a^	Ad2 ^a^
Ad4	Ad8	Ad9/Ad10 #		Ad5	Ad4	Ad7	Ad4 ^a^	Ad6 ^a^
				Ad9–Ad10 #		Ad9–Ad10 #	Ad5 ^a^	Ad4 ^a^
							Ad6 ^a^	Ad3 ^a^
							Ad10 ^a^	Ad5 ^a^
							Ad3 ^a^	Ad8 ^a^
							Ad8 ^a^	Ad10 ^a^
							Ad9 ^a^	Ad9 ^a^
							Ad12 ^b^	Ad13 ^b^
							Ad13 ^b^	Ad12 ^b^
							Ad14 ^b^	Ad14 ^b^

^†^—the efficiency of OP release was assessed by the yield of OPs or their fluoroanhydrides (ng/mL, pmol/mL pg/mL or % of recovery) in in vitro and in vivo experiments, by the distance between the functionally significant atoms, or by the value of the energy barrier in computational experiments; *—same efficiency for adducts; #—there was no division into stereoisomers in experiment; ^a^—BChE adducts; ^b^—adducts with Tyr150 of HSA; MM—molecular mechanics; AM1—semiempirical method AM1; hBChE—human BChE; mBChE—rhesus macaque BChE.

## Data Availability

The data presented in this study are available on request.

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
