# Peer review of "Structure-Dependent Mechanism of Organophosphate Release from Albumin and Butyrylcholinesterase Adducts When Exposed to Fluoride Ion: A Comprehensive In Silico Study"

_ijms, 2023, doi:10.3390/ijms241914819_

Round 1

Reviewer 1 Report

The manuscript "Structure-dependent mechanism of organophosphate release from albumin and butyrylcholinesterase adducts when exposed to fluoride ion: a comprehensive in silico study" uses computational modeling approaches to study the interactions of organophosphates (OPs) with human serum albumin (HSA) and butyrylcholinesterase (BChE). The authors performed molecular docking, molecular dynamics simulations, and quantum chemical calculations to model OP adduct formation on HSA and BChE, as well as fluoride-mediated cleavage of these adducts. The results provide insights into the structural factors that influence OP binding and reactivation of OP-inhibited BChE and HSA. Some key strengths of the study are:

  • The authors selected a diverse set of 11 OP adducts on BChE and 4 adducts on HSA to represent different OP chemical structures and stereoisomers. Modeling multiple adducts strengthens the conclusions.
  • The computational methods are appropriate for the research questions and system sizes. Docking was used for initial binding poses, MD provided dynamic information over nanosecond timescales, and QM methods calculated interaction energies.
  • The results generally agree with available experimental data on fluoride-mediated reactivation of BChE adducts, lending confidence to the computational approaches.
  • The proposed computational workflow could be useful for predicting reactivation potential of new OP agents.

Some minor aspects that could be improved:

  • More discussion of the limitations and assumptions of the modeling approaches is warranted. For example, OP-BChE aging is not accounted for.
  • Some key experimental studies are not referenced, such as kinetics of BChE reactivation by oximes. Comparison to reactivators besides fluoride would be informative.
  • Additional validation against experimental data would strengthen the conclusions, if available. For instance, binding free energies of OPs to BChE/HSA sites could be compared to measured values.
  • Figures depicting the full BChE/HSA structures with OP adducts would provide helpful visuals of the binding sites.

Author Response

a point-by-point response is provided in file attached 

Reviewer 2 Report

This is an excellent manuscript. It contains much more information than I would expect in a single paper. The authors certainly could have generated several papers from this material, but it is certainly not a problem that they have put it all into one publication.

The paper needs citations for the following text, beginning on line 43.

However, AChE is not the only target of OPs in the human organism. Butyrylcholinesterase (BChE, EC 3.1.1.8) has an active site similar to AChE, viz: the catalytic triad (Ser198, His438, and Glu325), the oxyanion center (Gly116, Gly117 and Ala199, binds the carboxylic oxygen of substrates and the phosphoryl oxygen of OPs), and the anionic center (includes Trp82 and binds ammonium cationic groups of substrates and inhibitors).

The chemical structure shown in Line 138
Ad17 o-hydroxybenzyl adduct

is not a phosphodiester or triester. even though it is in a table entitled ... 

Table 2. General formula and structures of the studied organophosphorus adducts on human serum albumin (HSA) 

Is the structure given there an error? It is not an organophosphorous compound.

For the text on line 288, 294, and 299

The complex of CBDP with His242 is stable.  

The complex of CBDP with Lys351 is unstable. 

The complex of CBDP with Tyr411 is stable for some period of the simulation.  

I assume that these designations of stable or unstable are derived from a computational method, not an experimental method. This should be made clear in the text. 

The following open-access software packages, VMD, GROMACS, and Autodock Vina are cited in the text by giving their source institution. This method of citation is insufficient. For instance, for Autodock Vina, the developers request that users of their software cite the following references and link those references in the text. Only in that way can the developers track the worldwide usage of their software by use of citation analyses. 

J. Eberhardt, D. Santos-Martins, A. F. Tillack, and S. Forli. (2021). AutoDock Vina 1.2.0: New Docking Methods, Expanded Force Field, and Python Bindings. Journal of Chemical Information and Modeling.

O. Trott, A. J. Olson, AutoDock Vina: improving the speed and accuracy of docking with a new scoring function, efficient optimization and multithreading, Journal of Computational Chemistry 31 (2010) 455-461

Such citations should be given for all software packages used in this paper and linked to the name of the software in the text. In this case, the authors do cite the second of the two papers that the developers of Autodock Vina, but not the first, and do not link it to each mention of the software. 

In Table 8. Efficiency of OP release 

What does the word "efficiency" mean? Does it have dimensional units?

I have no further comments. 

Author Response

(The authors gave the same response as above.)
